JCB — Journal of Cell Biology

## TOOLS

# A cytosolic IF1 reporter enables real-time visualization of severe mitochondrial membrane damage

Erqing Gao[1,3] ◉, Liwei Guan[2] ◉, Kehang Zhang[1] ◉, Shuze Qi[1] ◉, Jingxiu Xu[1,4] ◉, Jie Zhang[5] ◉, Tianyi Zhu[6] ◉, Bing Yang[6] ◉, Chonglin Yang[5] ◉, Eric Miska[4] ◉, Mao Zhang[7] ◉, and Suhong Xu[1,3,7] ◉

Maintenance of mitochondrial integrity is fundamental for cellular survival, yet how cells recognize catastrophic mitochondrial membrane damage remains unknown. Here, we identify MAI-1 as the first genetically encoded reporter of severe mitochondrial membrane damage. MAI-1 is a *Caenorhabditis elegans* homolog of the ATP synthase inhibitor IF1 that lacks a mitochondrial targeting sequence, resides in the cytosol under basal conditions, but rapidly and irreversibly translocates to severely damaged mitochondria within milliseconds. We validate MAI-1 across diverse injury paradigms and demonstrate that cytosolic IF1 variants from other species exhibit conserved damage-induced recruitment. Mechanistically, MAI-1 recruitment requires the presence of an intact ATP synthase complex. Using MAI-1 as a sensor, we uncover that these severely damaged mitochondria are cleared through the LGG-1–mediated, PINK1/PARKIN-independent lysosomal pathway. Together, our findings establish a powerful tool for visualizing severe mitochondrial membrane damage and reveal a surveillance mechanism dedicated to structural integrity control.

## Introduction

Mitochondria are central hubs for energy production, apoptosis regulation, and innate immunity, and their proper function critically depends on the structural integrity of the inner and outer membranes (Chan, 2006; Kraus et al., 2021). Severe damage to these membranes is catastrophic: compartmentalization collapses, ATP production ceases, and pro-apoptotic factors are released (Kroemer et al., 2007). Yet, unlike lysosomes and plasma membrane, where damage can be sensed by galectins that bind exposed glycans (Thurston et al., 2012; Jia et al., 2020), no dedicated mechanism has been identified for detecting physical membrane damage of mitochondria. Whether cells can actively sense and respond to mitochondrial membrane damage, therefore, remains a fundamental, unresolved question.

Current models of mitochondrial quality control are dominated by pathways such as PINK1/Parkin-mediated mitophagy, while eliminating depolarized organelles (Youle and Narendra, 2011; Wei et al., 2017). However, these pathways primarily detect bioenergetic dysfunction rather than membrane damage (Youle and Narendra, 2011; Narendra and Youle, 2024), leaving unclear how cells distinguish reversibly stressed mitochondria from those that are irreversibly compromised. Moreover, the lack of molecular probes that directly report membrane damage has hindered the mechanistic investigation of this process and obscured potentially distinct surveillance pathways.

Here, we introduce MAI-1, a genetically encoded reporter that specifically labels severely damaged mitochondria *in vivo* with millisecond (ms) resolution. MAI-1 is a *C. elegans* homolog of human ATP synthase inhibitor IF1. Unlike canonical mitochondrial proteins, MAI-1 lacks a mitochondrial targeting sequence (MTS) and is cytosolic under basal conditions but rapidly and irreversibly translocates to mitochondria upon membrane damage. We show that MAI-1 selectively marks catastrophic structural damage rather than reversible depolarization, functions across species, and enables long-term tracking of damaged organelles in living animals. Using this tool, we uncover that membrane damage-marked mitochondria undergo lysosome-dependent degradation mediated by LGG-1, but independent of canonical PINK1/PARKIN mitophagy, revealing a previously unrecognized branch of mitochondrial quality control engaged by catastrophic membrane injury.

[1]Center for Stem Cell and Regenerative Medicine, Zhejiang University-University of Edinburgh Institute, Department of Burn and Wound Repair of the Second Affiliated Hospital, Zhejiang University School of Medicine, Hangzhou, China;   [2]Hangzhou Medical College, Hangzhou, China;   [3]Biomedical Sciences, College of Medicine and Veterinary Medicine, University of Edinburgh, Edinburgh, UK;   [4]Department of Biochemistry, University of Cambridge, Cambridge, UK;   [5]School of Life Sciences, Yunnan University, Yunnan, China;   [6]Life Sciences Institute, Zhejiang University, Hangzhou, China;   [7]Zhejiang Key Laboratory of Trauma, Burn, and Medical Rescue, Hangzhou, China.

Correspondence to Suhong Xu: shxu@zju.edu.cn.

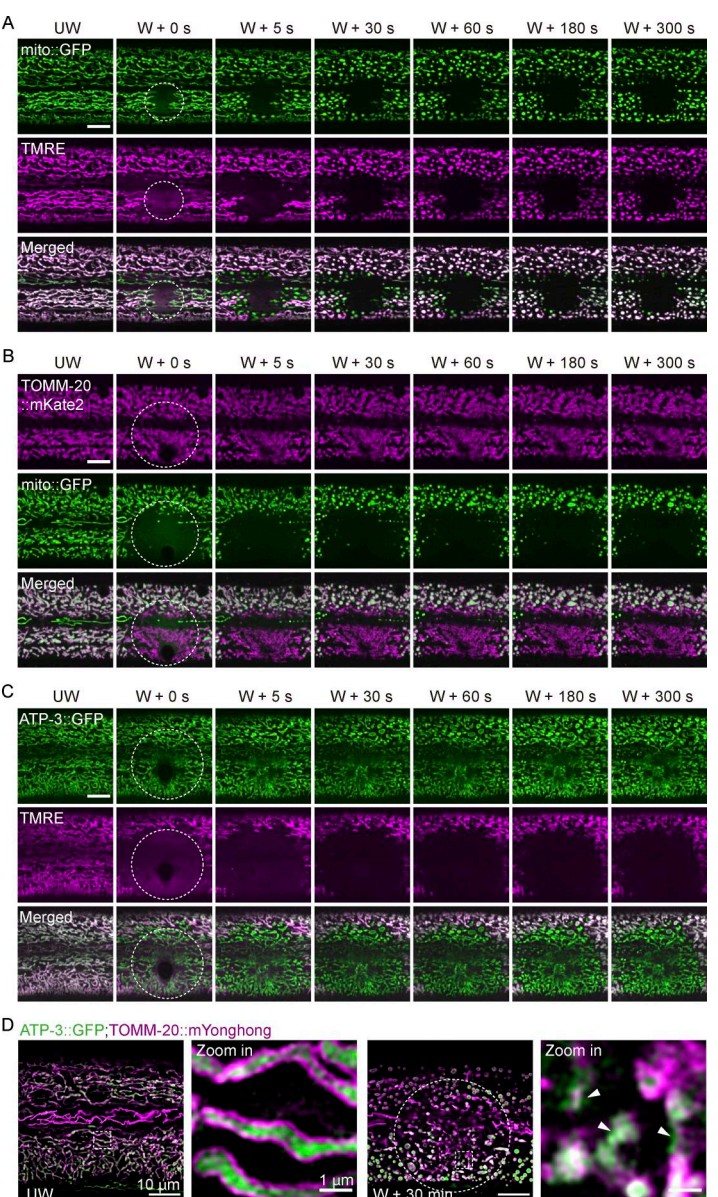

**Figure 1. Laser injury induces mitochondrial membrane permeabilization. (A)** Representative time-lapse confocal images of mito::GFP and TMRE before and after laser wounding. Strain: Pcol-19-mito::GFP (*zjuSi197*). Scale bar, 10 μm. **(B)** Representative time-lapse confocal images of TOMM-20::mKate2 and mito::GFP before and after laser wounding. Strain: Phyp-7-TOMM-20::mKate2 (*zjuSi126*); Pcol-19-mito::GFP (*zjuSi197*). Scale bar, 10 μm. **(C)** Representative time-lapse confocal images of ATP-3::GFP and TMRE before and after laser wounding. Strain: ATP-3::GFP (KI, *zju133*). Scale bar, 10 μm. **(D)** Representative single-plane HIS-SIM images of ATP-3::GFP and TOMM-20::mYonghong before and after laser wounding. ROI, 1660 × 1660 pixels. Strains: ATP-3::GFP(*zju133*); Pcol-19-TOMM-20::mYonghong (*zjuSi577*). Scale bar, 10 and 1 μm (zoom in). The arrowheads indicate disruption of TOMM-20::mYonghong and ATP-3::GFP signals. **(A–D)** The white dashed circle indicates the wound site. UW, unwounded; W, wounded.

## Results

### Laser injury induces mitochondrial membrane permeabilization

To define the immediate fate of mitochondria under catastrophic cellular stress, we performed precision laser wounding of the *C. elegans* epidermis and monitored mitochondrial response using complementary markers (Fu et al., 2020). Within seconds of injury, mitochondrial matrix marker (mito::GFP) and membrane potential (ΔΨm) indicator tetramethyl rhodamine ethyl ester (TMRE) dissipated at the wound site (Fig. 1 A, Fig. S1 A, and Video 1), indicating rapid loss of matrix contents and ΔΨm. In contrast, the outer mitochondrial membrane (OMM) marker TOMM-20::GFP remained detectable, with its fluorescence intensity modestly reduced (Fig. 1 B; Fig. S1, A and B; and Video 2). Furthermore, quantitative analysis of GFP fluorescence revealed an increase in cytosolic fluorescence signal at the wound site in mito::GFP animals, but not in animals expressing diffusely distributed GFP, compared with unwounded controls (Fig. S1, C and

D). Together, these observations suggest that acute laser injury causes the release of mitochondrial contents while largely preserving membrane scaffolds.

To distinguish inner versus outer membrane behavior, we wounded animals expressing the inner mitochondrial membrane protein ATP-3::GFP together with TMRE staining. Laser injury triggered immediate loss of TMRE signal, whereas ATP-3::GFP persisted at the wound site (Fig. 1 C; Fig. S1, A and E; and Video 2). This dissociation between membrane potential loss and persistence of membrane-associated proteins indicates that mitochondrial damage leads to loss of matrix components while leaving remnant membrane scaffolds.

Super-resolution imaging using structured illumination microscopy (SIM) revealed that TOMM-20::mYonghong forms a continuous outer membrane signal that encircles ATP-3::GFP in intact mitochondria before wounding (Fig. 1 D). After laser wounding, TOMM-20::mYonghong signal at the damage site appeared locally discontinuous and fragmented, while

remaining detectable (Fig. 1 D), consistent with observations from spinning-disk confocal imaging (Fig. S1 F). Notably, over the subsequent 2–24 h, the TOMM-20 signal at the wound site progressively diminished, whereas neighboring mitochondria remained intact (Fig. S1 G), suggesting selective clearance of damaged mitochondria. Together, these observations demonstrate that laser injury induces localized mitochondrial membrane permeabilization, characterized by loss of membrane potential and release of matrix contents, while overall membrane architecture remains largely preserved. This raises the question of how damaged mitochondria are specifically recognized *in vivo*.

## Discovery of a genetically encoded reporter for mitochondrial membrane damage

Traditional voltage-sensitive dyes report depolarization but cannot discriminate reversible bioenergetic fluctuations from catastrophic structural failure (Scaduto and Grotyohann, 1999). To address this gap, we sought a genetically encoded protein that remains cytosolic under basal conditions yet selectively relocalizes to mitochondria only upon damage. Guided by this requirement, we focused on IF1-family ATP synthase inhibitors (Fig. 2 A). Notably, *C. elegans* MAI-1 lacks a predicted MTS and was previously identified as a rapidly wound-induced transcript (Hu et al., 2023), whereas MAI-2 contains an MTS and is constitutively mitochondrial (Fig. S2 A) (Fernández-Cárdenas et al., 2017; Ichikawa et al., 2006). MAI-1 is broadly expressed in epidermis, neurons, and muscles (Fig. S2 B). In uninjured animals, MAI-1::GFP displayed diffuse cytosolic with no discernible localization to any organelles (Fig. 2, B and C). Strikingly, immediately after laser wounding, MAI-1::GFP rapidly and specifically accumulated on tubular and network-like structures resembling mitochondria (Fig. 2 B and Video 3), whereas diffuse cytosolic GFP did not show any relocalization (Fig. S2 C).

To confirm the identity of these structures, we performed co-localization analysis with a mitochondrial marker. Before wounding, MAI-1::GFP was spatially excluded from mitochondria labeled with TMRE or mito::mKate2 (Fig. 2 C and Fig. S2 D). After wounding, MAI-1::GFP selectively accumulated at sites where mito::mKate2 and TMRE signals were lost (Fig. 2 C and Fig. S2 D), consistent with recruitment to damaged mitochondria. This translocation occurred extremely rapidly, within ~200 milliseconds (ms) of injury (Fig. 2 D). Co-imaging with the TOMM-20::mKate2 further showed that MAI-1::GFP progressively enriched on TOMM-20::mKate2-positive mitochondrial remnants at the injury site (Fig. 2 E and Video 4) and was not photoconverted from the mKate2 signal (Fig. S2 F). Quantitative colocalization analysis confirmed strong spatial coincidence between MAI-1 and TOMM-20 or ATP-3 after injury (Fig. 2, F and G). Importantly, MAI-1 translocation was independent of Tag orientation (N- or C-terminal) or promoter context (Fig. S2, D and E). Together, these results establish MAI-1 as a genetically encoded reporter that specifically labels severely damaged mitochondria *in vivo*.

## MAI-1 is a conserved IF1 family protein that translocates to damaged mitochondria without an MTS

Sequence analysis revealed that MAI-1 is a highly conserved member of the IF1 family despite lacking a canonical MTS (Fig. 3, A and B). While most vertebrates encode a single IF1 protein that contains an MTS, many nematode species, as well as *Drosophila* and zebrafish, encode two IF1 paralogs: one contains an MTS (MAI-2-like) and one lacks an MTS (MAI-1–like) (Fig. S2 G).

To test whether damage-induced recruitment is a conserved property of cytosolic IF1 (cIF1) proteins, we generated MTS-deleted variants of human IF1 and *C. elegans* MAI-2 (hereafter "cytosolic IF1," cIF1). Full-length IF1 localized constitutively to mitochondria (Fig. S2 H), whereas cIF1 variants were diffusely cytosolic under basal conditions (Fig. 3, C and D). Remarkably, both human and worm cIF1 rapidly translocated to damaged mitochondria upon laser wounding, mimicking MAI-1 behavior (Fig. 3, C and D; and Video 5). MAI-1 is a small 88-amino acid protein. Truncation of either N or C terminus abolished damage-induced recruitment (Fig. 3, E–G), suggesting that the intact protein is required for its targeting function. Collectively, these data suggest that IF1 family proteins possess an intrinsic capacity to associate with mitochondria upon severe membrane damage when present in the cytosol.

## MAI-1 reports the mitochondrial damage induced by diverse stresses

We next asked whether MAI-1 detects damage beyond laser-induced injury. Mechanical puncture by microinjection needle and ultrasound-induced shear stress both triggered MAI-1::GFP recruitment to TOMM-20–labeled mitochondria (Fig. 4 A and Fig. S3 A). Acute cold stress causes a complete loss of mito::GFP fluorescence signal, while TOMM-20::mKate2 remained associated with damaged mitochondria (Fig. S3 B) and similarly induced MAI-1::GFP accumulation on TOMM-20::mKate2-labeled mitochondria (Fig. 4 A). Together, these results suggest that diverse physical stresses induce mitochondrial membrane damage and trigger MAI-1 translocation *in vivo*.

Because mitochondria are highly dynamic organelles undergoing continual fusion or fission (Chan, 2006), we asked whether perturbation of mitochondrial dynamics is sufficient to trigger MAI-1 recruitment. In mutants defective in mitochondrial fission (*drp-1*/DRP1), fusion (*fzo-1*/MFN1/2; *eat-3*/OPA1), crista organization (*chch-3*/CHCHD3), or mitophagy regulators (*pink-1*/PINK1), MAI-1::GFP remained diffusely cytosolic under basal conditions and translocated to mitochondria only after laser injury (Fig. S3 C). By contrast, in mutants with severe mitochondrial metabolism defects (*argn-1* and *aass-1*), a subset of MAI-1::GFP was constitutively associated with mitochondria even in the absence of exogenous injury (Fig. 4 B and Fig. S3 D), consistent with reported mitochondrial membrane damage in these backgrounds (Tang et al., 2020; Zhou et al., 2019). Together, these observations suggest that MAI-1 is not recruited by perturbations in mitochondrial dynamics per se, but can recognize mitochondria undergoing endogenous membrane compromise under physiological conditions.

To establish a direct link between mitochondria-specific damage and MAI-1 translocation, we employed an optogenetic

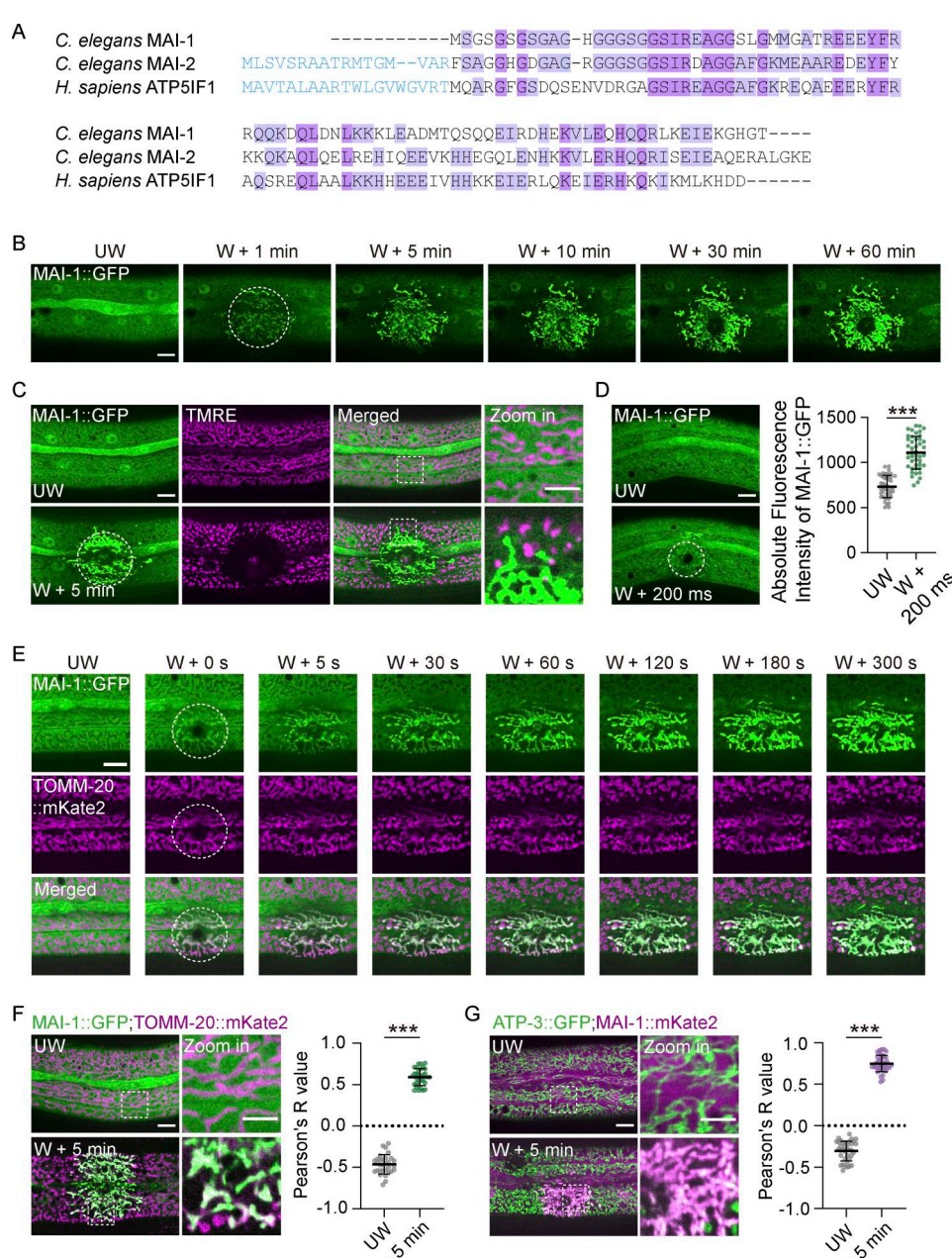

Figure 2. **Discovery of a genetically encoded mitochondrial damage reporter. (A)** Amino acid sequences alignment of *C. elegans* MAI-1, MAI-2, and human ATP5IF1. **(B)** Representative time-lapse confocal images of MAI-1::GFP before and at different time points (1–60 min) after laser wounding. Strain: P*col-19*-MAI-1::GFP(*zjuSi510*). Scale bar, 10 μm. **(C)** Representative time-lapse confocal images of MAI-1::GFP;TMRE before and after laser wounding. Strains: P*col-19*-MAI-1::GFP(*zjuSi510*). Scale bar, 10 and 5 μm (zoom in). **(D)** Left, representative confocal images of MAI-1::GFP before and 200 ms after laser wounding (first frame taken after laser wounding). Frame rate: 200 ms; exposure time: 100 ms. Strain: P*col-19*-MAI-1::GFP(*zjuSi510*). Scale bar, 10 μm. Right, quantitation of absolute MAI-1::GFP intensity before and after wounding to the left images. Multiple mitochondria within an ROI (6.9 μm × 6.9 μm) at the wound site were measured. *n* represents the number of ROI analyzed. Statistical analysis was performed using the Mann–Whitney test. Data are mean ± SD (*n* = 45 and 45, respectively); ***, P < 0.001. **(E)** Representative time-lapse confocal images of MAI-1::GFP; TOMM-20::mKate before and at different time points after laser wounding. Strains: P*hyp-7*-TOMM-20::mKate2(*zjuSi126*); P*col-19*-MAI-1::GFP(*zjuSi510*). Scale bar, 10 μm. **(B–E)** The white dashed circle indicates the wound site. UW, unwounded; W, wounded. **(F)** Left, representative confocal images of MAI-1::GFP and TOMM-20::mKate2 before and after laser wounding at the wound site. Strains: P*hyp-7*-TOMM-20::mKate2 (*zjuSi126*); P*col-19*-MAI-1::GFP (*zjuSi510*). Scale bar, 10 and 5 μm (zoom in). Right, Pearson's colocalization analysis of MAI-1::GFP and TOMM-20::mKate2 before and after wounding to the left images. Multiple mitochondria within an ROI (13.8 μm × 13.8 μm) at the wound site were measured. *n* represents the number of ROI analyzed. Statistical analysis was performed using the Mann–Whitney test. Data are mean ± SD (*n* = 31 and 32, respectively); ***, P < 0.001. **(G)** Representative confocal images of ATP-3::GFP and MAI-1::mKate2 before and after laser wounding at the wound site. Strains: ATP-3::GFP (*zju133*); P*col-19*-MAI-1::mKate2 (*zjuSi510*). Scale bar, 10 and 5 μm (zoom in). Right, Pearson's colocalization analysis of MAI-1::mKate2 and ATP-3::GFP before and after wounding to the left images. Multiple mitochondria in an ROI (13.8 μm × 13.8 μm) at the wound site were measured. *n* represents the number of ROI analyzed. Statistical analysis was performed using the Mann–Whitney test. Data are mean ± SD (*n* = 37 and 38, respectively); ***, P < 0.001.

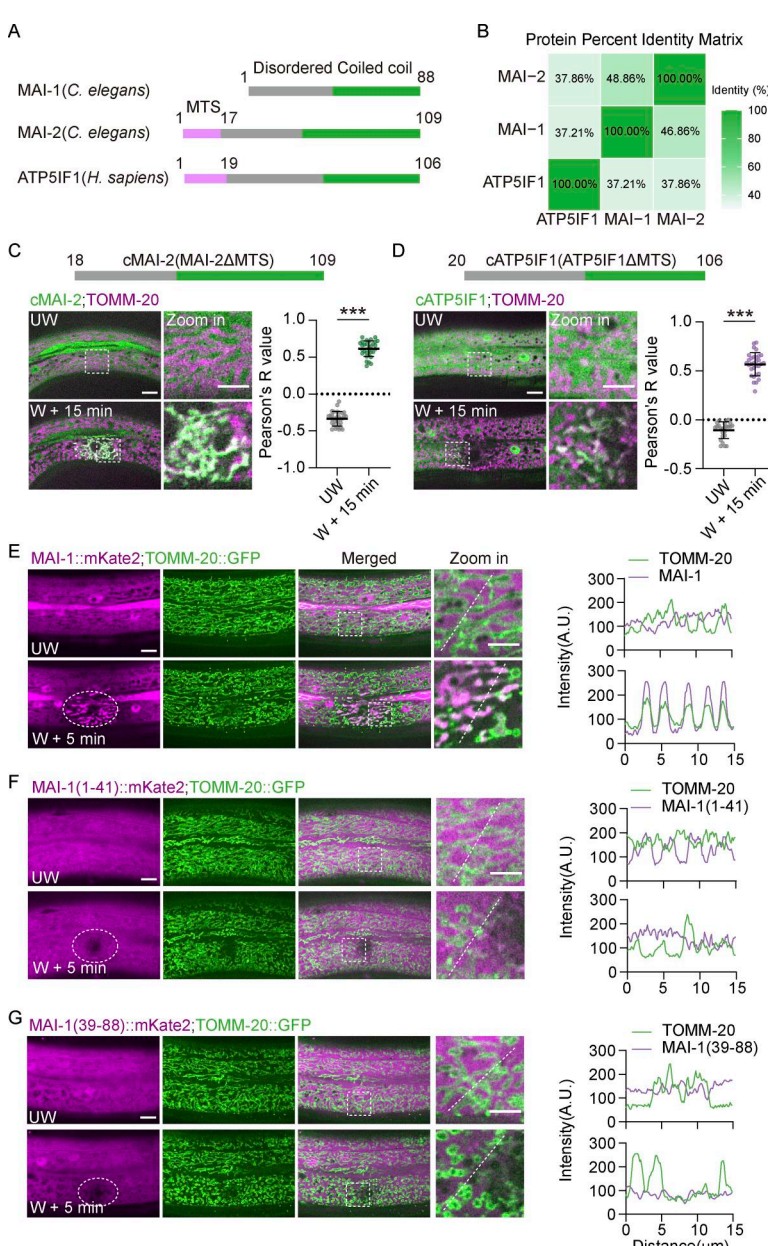

Figure 3. **cIF1 translocates to damaged mitochondria.** **(A)** Domain architecture of *C. elegans* MAI-1, MAI-2, and human IF1. **(B)** Protein percent identity matrix analysis among *C. elegans* MAI-1, MAI-2, and human IF1. Values represent pairwise percent identity. **(C)** Representative confocal images of cytosolic MAI-2-GFP fusion protein localization before and after laser wounding. Strains: P*hyp-7*-TOMM-20::mKate2(*zjuSi126*); P*col-19*-MAI-2(ΔMTS)(17–109 aa)::GFP(*zjuEx3449*). Scale bar: 10 and 5 µm (zoom in). Right, Pearson's colocalization analysis of cytosolic MAI-2-GFP and TOMM-20::mKate2 before and after wounding, as shown in the left images. Statistical analysis was performed using the Mann–Whitney test. Data are mean ± SD (*n* = 31 ROIs); ***, P < 0.001. **(D)** Representative confocal images of cIF1-GFP fusion protein localization before and after laser wounding. Strains: P*hyp-7*-TOMM-20::mKate2(*zjuSi126*); P*col-19*-IF1(ΔMTS)(20–106 aa)::GFP (*zjuEx3447*). Scale bar: 10 and 5 µm (zoom in). Right, Pearson's colocalization analysis of cIF1-GFP and TOMM-20::mKate2 before and after wounding to the left images. Statistical analysis was performed using the Mann–Whitney test. Data are mean ± SD (*n* = 25 and 31 ROIs, respectively); ***, P < 0.001. **(E–G)** Representative images of MAI-1 truncation mutants before and after wounding. Strains: P*col-19*-TOMM-20::GFP(*zjuSi42*); P*col-19*-MAI-1::mKate2(*zjuEx3209*), P*col-19*-MAI-1(1–41)::mKate2(*zjuEx3212*), and P*col-19*-MAI-1(39–88)::mKate2(*zjuEx3461*). Scale bar, 10 and 5 µm (zoom in). Right, quantification of fluorescence intensity along 15 µm lines across the unwound area and wound site. The white dashed circle indicates the wound site. UW, unwounded; W, wounded. The white dashed lines indicate the analysis region.

tool using TOMM-20::miniSOG to generate sustained singlet oxygen at the OMM upon blue-light illumination (Xu and Chisholm, 2016; Qi et al., 2012). In the absence of illumination, MAI-1::GFP remained diffusely cytosolic (Fig. 4 C). Upon blue-light exposure, TOMM-20::mKate2-labeled mitochondria progressively shrank and fragmented, accompanied by a time-dependent local loss of fluorescence (Fig. 4 C), consistent with severe mitochondrial membrane damage that can lead to cell or organismal lethality (Qi et al., 2012; Xu and Chisholm, 2016). Concomitantly, MAI-1::GFP robustly translocated to the damaged mitochondria (Fig. 4 C). Super-resolution SIM imaging further revealed that MAI-1::GFP localized within the TOMM-20::mKate2-outlined fragmented mitochondrial remnants following damage (Fig. 4 D). Together, these results establish MAI-1 as a universal reporter of severe mitochondrial membrane damage induced by diverse cellular stress.

## MAI-1 distinguishes mitochondrial damage from reversible depolarization

We next asked whether MAI-1 recruitment simply reflects mitochondrial depolarization or instead detects membrane damage. Live imaging revealed that MAI-1 translocated only to mitochondria with sustained loss of membrane potential, as indicated by irreversible TMRE disappearance (Fig. 5 A (a1) and Video 6). In contrast, neighboring mitochondria that underwent transient TMRE loss followed by full recovery did not recruit MAI-1 (Fig. 5 A (a2) and Video 6). Quantitation analysis confirmed that MAI-1::GFP progressively accumulated on mitochondria that permanently lost TMRE, but not on mitochondria in which TMRE fluorescence transiently decreased and recovered (Fig. 5, A–C). Targeted laser ablation of individual tubular mitochondria yielded the same outcome: MAI-1::GFP progressively accumulated on fragmented organelles displaying

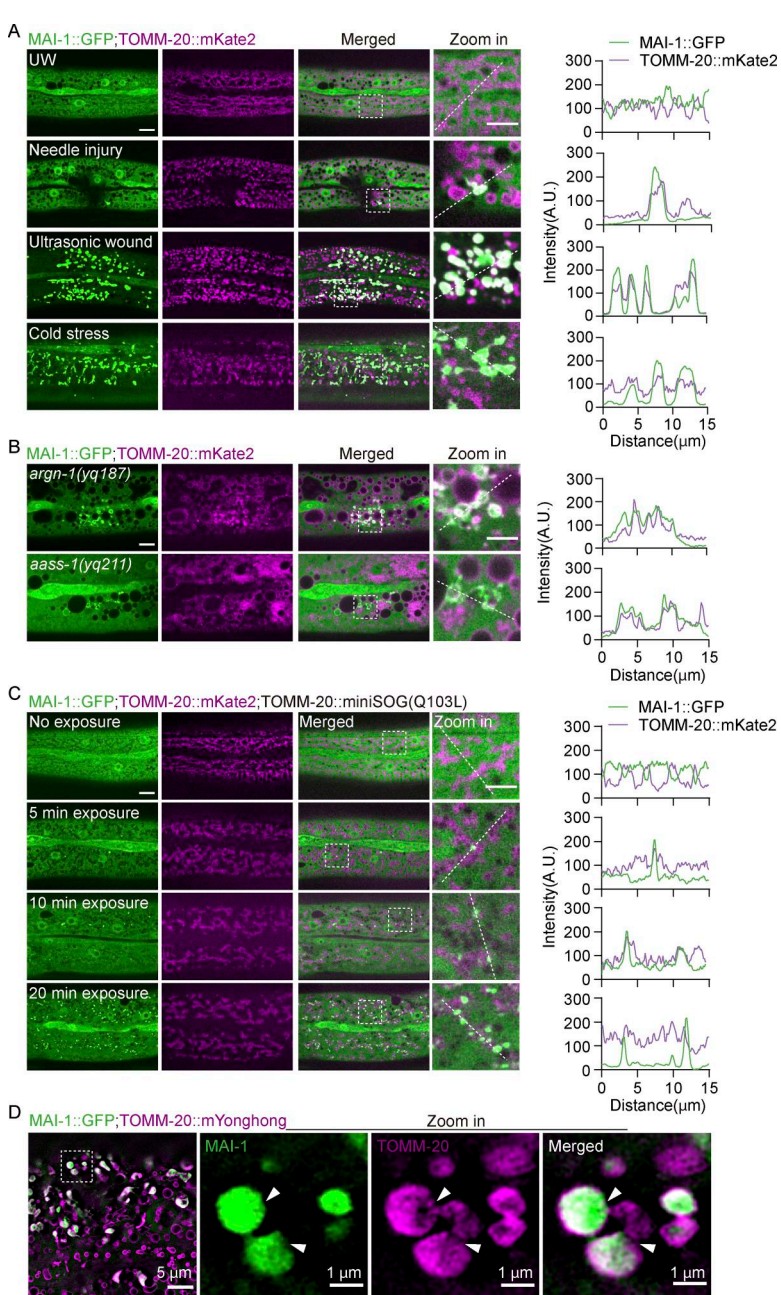

Figure 4. **MAI-1 reports mitochondrial damage induced by diverse stresses. (A)** Left, representative confocal image of MAI-1::GFP and TOMM-20::mKate2 in the epidermal hyp7 before and after mechanical puncture, ultrasonic, and cold shock (−20°C) stress. Right, quantification of fluorescence intensity along 15 μm lines across the wound site. Strains: Phyp-7-TOMM-20::mKate2 (zjuSi126); Pcol-19-MAI-1::GFP (zjuSi510). The white dashed lines indicate the analysis region. Scale bar, 10 and 5 μm (zoom in). **(B)** Left, representative confocal image of MAI-1::GFP and TOMM-20::mKate2 in aass-1 and argn-1 mutant. Enlarged and damaged mitochondria colocalize with MAI-1::GFP. Right, quantification of fluorescence intensity along 15 μm lines across the wound site. Strains: aass-1 (yq211), argn-1 (yq187); Phyp-7-TOMM-20::mKate2(zjuSi126); Pcol-19-MAI-1::GFP (zjuSi510). The white dashed lines indicate the analysis region. Scale bar, 10 and 5 μm (zoom in). **(C)** Left, representative confocal images of MAI-1::GFP and TOMM-20::mKate2 after optogenetic mitochondrial damage. The entire NGM plate was uniformly illuminated with blue light. 2 HZ blue-light illumination of TOMM-20::miniSOG-induced sustained mitochondrial damage. Right, quantification of fluorescence intensity along 15 μm lines across the wound site. Strains: Phyp-7-TOMM-20::mKate2 (zjuSi126); Pcol-19-MAI-1-GFP (zjuSi510); Pcol-19-TOMM-20::miniSOG (zjuEx3393). The white dashed lines indicate the analysis region. Scale bar, 10 and 5 μm (zoom in). **(D)** Representative single-plane HIS-SIM images of MAI-1::GFP and TOMM-20::mYonghong after ultrasonically induced damage. The white arrowheads indicate membrane damage. ROI, 1024 × 1024 pixels. Strains: Pcol-19-TOMM-20::mYonghong(zjuSi577); Pcol-19-MAI-1::GFP (zjuSi510). Scale bar, 5 and 1 μm (zoom in).

irreversible TMRE loss, whereas mitochondria that displayed transient decreased and recovered membrane potential did not recruit MAI-1 (Fig. 5, D–F and Video 7). Thus, MAI-1 specifically labels mitochondria that undergo catastrophic and irreversible damage, rather than reversible bioenergetic fluctuations.

To directly decouple membrane depolarization from membrane damage, we applied pharmacological inhibitors that abolish ΔΨm without physically breaching mitochondrial membranes. Treatment with FCCP, rotenone, or oligomycin eliminated TMRE staining and induced mitochondrial fragmentation, but did not trigger MAI-1::GFP recruitment (Fig. 5 G; and Fig. S4, A and B). These results demonstrate that loss of ΔΨm alone is not sufficient to trigger MAI-1 translocation. Thus, MAI-1 selectively labels mitochondria that undergo catastrophic and irreversible membrane damage, rather than reversible bioenergetic fluctuations.

## MAI-1 recruitment to damaged mitochondria requires ATP synthase

We next sought to define the mechanism by which MAI-1 recognizes damaged mitochondria. Human IF1 binds the interface between the α and β subunits of ATP synthase (Esparza-Moltó et al., 2017). Consistent with this, MAI-1::mKate2 exhibited strong colocalization with the ATP synthase subunit ATP-3::GFP at mitochondrial injury sites (Fig. 6 A). Triple color colocalization of TOMM-20::mYonghong, ATP-3::GFP, and MAI-1::BFP further indicates that MAI-1 associates with inner membrane structure exposed within damaged outer membrane remnants (Fig. 6 B).

To test whether ATP synthase is required for MAI-1 recruitment, we depleted essential ATP synthase subunits by RNAi and quantified MAI-1::GFP accumulation at laser-damaged

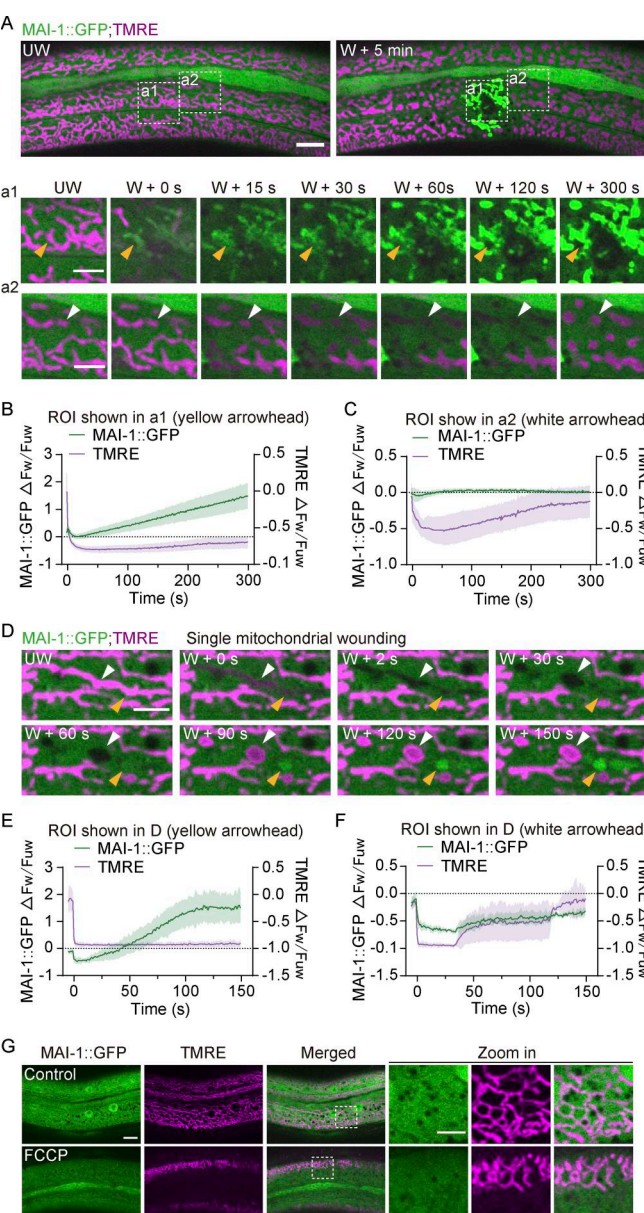

Figure 5. **MAI-1 distinguishes damage from reversible membrane depolarization. (A)** Time-lapse images showing MAI-1::GFP and TMRE after large laser wounding. MAI-1::GFP only translocated to irreversibly depolarized mitochondria (a1) but not recovered mitochondria (a2). The yellow arrowhead (a1) indicates irreversibly depolarized mitochondria. The white arrowhead (a2) indicates recovered mitochondria. Strains: P$col$-19-MAI-1::GFP ($zjuSi510$). Scale bar, 10 and 5 μm (zoom in). UW, unwounded; W, wounded. **(B)** Line graph showing the intensity change (ΔFw/Fuw) of MAI-1::GFP and TMRE (yellow arrowhead in panel D) in the large laser-wounded site at different time points after wounding (a1). Multiple mitochondria in an ROI (13.8 μm × 13.8 μm) at the wound site were measured. $n$ represents the number of ROIs. Data are mean ± SD, $n$ = 3. **(C)** Line graph showing the intensity change (ΔFw/Fuw) of MAI-1::GFP and TMRE (white arrowhead in panel D) near the large laser-wounded site at different time points after wounding (a2). Data are mean ± SD, $n$ = 3. **(D)** Representative confocal time-lapse images of translocation of MAI-1::GFP and recovery of TMRE at the small laser wound site in 150 s. The yellow arrowhead indicates irreversibly depolarized mitochondria, and the white arrowhead indicates recovered mitochondria. Strains: P$col$-19-MAI-1::GFP ($zjuSi510$). Scale bar, 5 μm. **(E)** Line graph showing the intensity change (ΔFw/Fuw) of MAI-1::GFP and TMRE (yellow arrowhead in panel D) at different time points after wounding. A

single fragmented mitochondrion was measured. Data are mean ± SD, $n$ = 5. **(F)** Line graph showing the intensity change (ΔFw/Fuw) of MAI-1::GFP and TMRE (white arrowhead in panel D) at different time points after wounding. A single fragmented mitochondrion was measured. Data are mean ± SD, $n$ = 3. **(G)** Representative confocal images of MAI-1::GFP after FCCP treatment (15 μM, 2 h). TMRE was abolished, but MAI-1 remains cytosolic. Strains: P$col$-19-MAI-1::GFP ($zjuSi510$). Scale bar: 10 and 5 μm (zoom in).

mitochondria. Remarkably, we found that knockdown of $atp$-1 or $atp$-3 almost completely blocked MAI-1::GFP accumulation at wound sites at both 1 and 5 min after damage compared with controls (Fig. 6, C and D). ATP-1 and ATP-3 are the $C. elegans$ orthologs of human ATP synthase components ATP5F1A and ATP5PO, respectively. Together, these results indicate that the presence of an intact ATP synthase complex is required for efficient MAI-1 recruitment after wounding.

## MAI-1–marked mitochondria undergo autophagy-dependent degradation

We next investigate the fate of MAI-1–marked mitochondria following injury. Time-lapse imaging revealed that MAI-1 remained stably associated with damaged mitochondria for several hours before both mitochondrial and MAI-1 signals disappeared (Fig. 7 A), indicating subsequent organelle clearance. To determine whether this clearance depends on the canonical mitophagy pathway, we examined the $pink$-1($ok3538$) and $pdr$-1($gk448$) loss-of-function mutants. Strikingly, MAI-1–positive mitochondria were removed with kinetics similar to WT in both mutants (Fig. 7 B and Fig. S5 A), indicating that clearance occurs independently of the canonical PINK1/PARKIN-independent pathway.

By contrast, in autophagy-deficient mutants $lgg$-1($tm3489$) and $atg$-3($bp412$), MAI-1–marked damaged mitochondria remnants persisted at the wound site for over 24 h (Fig. 7, B and C), demonstrating that autophagy is essential for the removal of damaged mitochondria. Consistently, LGG-1 (the $C. elegans$ ATG8/LC3 homology) (Kovacs and Zhang, 2010) was robustly recruited to mitochondrial injury sites, where it rapidly fused with MAI-1–marked organelles (Fig. 7 D and Video 8). Together, these findings suggest that damaged mitochondria are removed through an LGG-1/LC3-dependent autophagy pathway that is molecularly distinct from canonical PINK1/PARKIN-dependent mitophagy.

## Lysosomes engulf MAI-1–marked mitochondria after injury

Finally, we examined the terminal fate of MAI-1–marked mitochondria. Because autophagic removal ultimately requires lysosome degradation (Narendra and Youle, 2024; Youle and Narendra, 2011), we co-expressed MAI-1::GFP with the lysosomal lumen marker NUC-1::mCherry. During the first hour following injury, MAI-1–positive mitochondria remain spatially distinct from the lysosome (Fig. S5 B). By 2 h after wounding, NUC-1::mCherry progressively associated with MAI-1–marked fragments, followed by the disappearance of the MAI-1::GFP signal (Fig. 8 A). Targeted laser wounding on individual mitochondria further revealed that MAI-1–marked damaged mitochondria fragments became enveloped by lysosome structures,

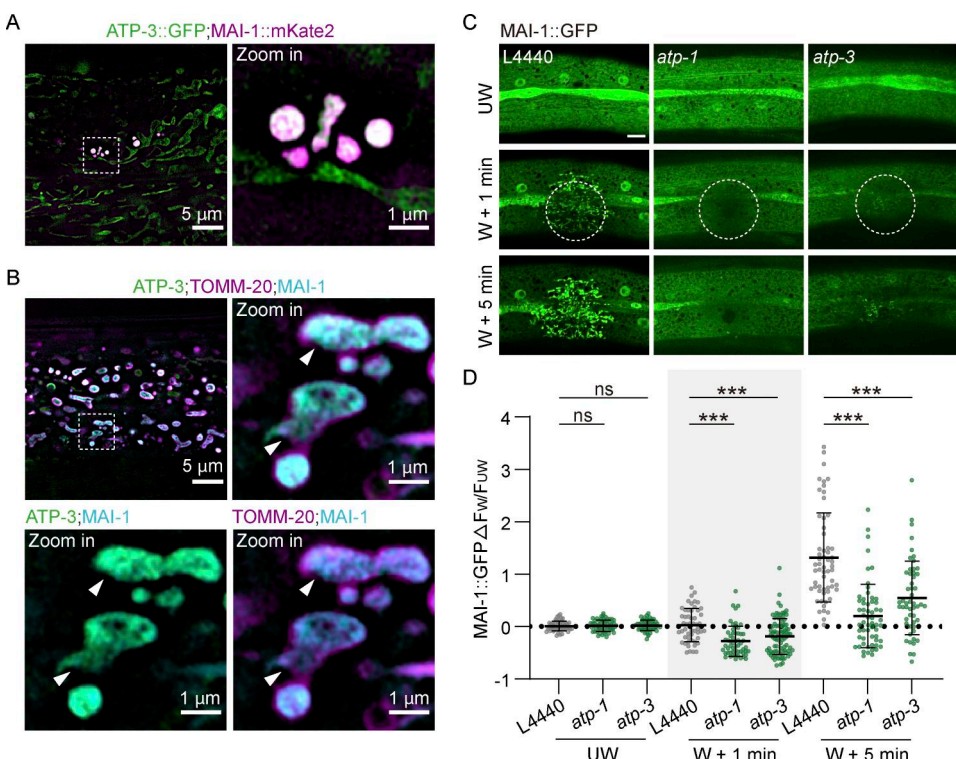

Figure 6. **MAI-1 recruitment to damaged mitochondria requires ATP synthase. (A)** Representative HIS-SIM single-plane images of ATP-3::GFP and MAI-1::mKate2 before and after ultrasonically induced damage. ROI, 1024 × 1024 pixels. Strains: ATP-3::GFP (*zju133*); P*col-19*-MAI-1::mKate2 (*zjuSi594*). Scale bar, 5 and 1 μm (zoom in). **(B)** Representative HIS-SIM single-plane images of ATP-3::GFP, TOMM-20::mYonghong, and MAI-1::BFP before and after ultrasonically induced damage. The white arrowhead indicates the membrane damage. ROI, 1024 × 1024 pixels. Strains: ATP-3::GFP (*zju133*); P*col-19*-TOMM-20::mYonghong (*zjuSi577*); P*col-19*-MAI-1::BFP (*zjuSi539*). Scale bar, 5 and 1 μm (zoom in). **(C)** Representative confocal images of MAI-1::GFP under L4440 (empty vector), *atp-1*, and *atp-3* RNAi treatment 1 and 5 min after laser wounding. Strains: P*col-19*-MAI-1::GFP (*zjuSi510*). The white dashed circle indicates the wound site. UW, unwounded; W, wounded. Scale bar, 10 μm (zoom in). **(D)** Quantitation analysis of MAI-1::GFP intensity change at the injury site. *n* represents the number of wounds. Statistical analysis was performed using one-way ANOVA. Error bars represent the mean value ± SD (*n* = 50, 57, 54, 52, 56, 87, 59, 58, and 52 from left to right); ***, P < 0.001, compared with L4440.

after which the MAI-1 signal was lost (Fig. 8 B and Video 9). These observations suggest MAI-1 marks damaged mitochondria for selective lysosomal degradation.

We next performed triple labeling of TOMM-20::mYonghong, NUC-1::mNeonGreen, and MAI-1::BFP, which revealed spatial convergence of mitochondria, lysosomes, and MAI-1 signals after injury, as confirmed by line-scan analysis (Fig. 8 C). Because the acidic lysosomal lumen quenches GFP fluorescence, we further imaged lysosomal membranes labeled with SCAV-3::GFP together with MAI-1::mKate2 in time-lapse recordings, which showed lysosomal membranes approaching and enveloping MAI-1–positive mitochondria following laser damage (Fig. 8 D, Fig. S5 C, and Video 10). Together, these results suggest that lysosomes directly engage and engulf MAI-1–marked mitochondria during injury-induced mitochondrial clearance.

## Discussion

This study discovers a previously unrecognized branch of mitochondrial quality control dedicated to sensing catastrophic membrane damage. By identifying a cIF1 homolog as a genetically encoded damage sensor, we provide a molecular solution to the long-standing problem: how cells discriminate irreversible

membrane structural collapse from reversible bioenergetic stress.

A key conceptual advance is the clear decoupling of mitochondrial membrane integrity surveillance from the membrane potential monitoring. The canonical PINK1/PARKIN pathway exemplifies functional quality control, in which loss of ΔΨm serves as a proxy for mitochondrial dysfunction (Youle and Narendra, 2011). In contrast, MAI-1 remains cytosolic during pharmacological depolarization yet translocates within seconds to damaged mitochondria, revealing a parallel system dedicated to structural damage. Together, these dual surveillance mechanisms equip cells to discriminate between transient metabolic perturbations and irreversible breaches that risk release of pro-apoptotic and pro-inflammatory factors, ensuring damage-appropriate response.

Our data further indicate that MAI-1 recruitment requires the presence of an intact ATP synthase complex, supporting ATP synthase as a primary binding target that mediates damage-induced association of MAI-1 with damaged mitochondria. This suggests a mechanistic model in which inner membrane disruption exposes ATP synthase interfaces normally inaccessible to cytosolic proteins, thereby licensing selective recognition of catastrophically damaged organelles. Whether additional

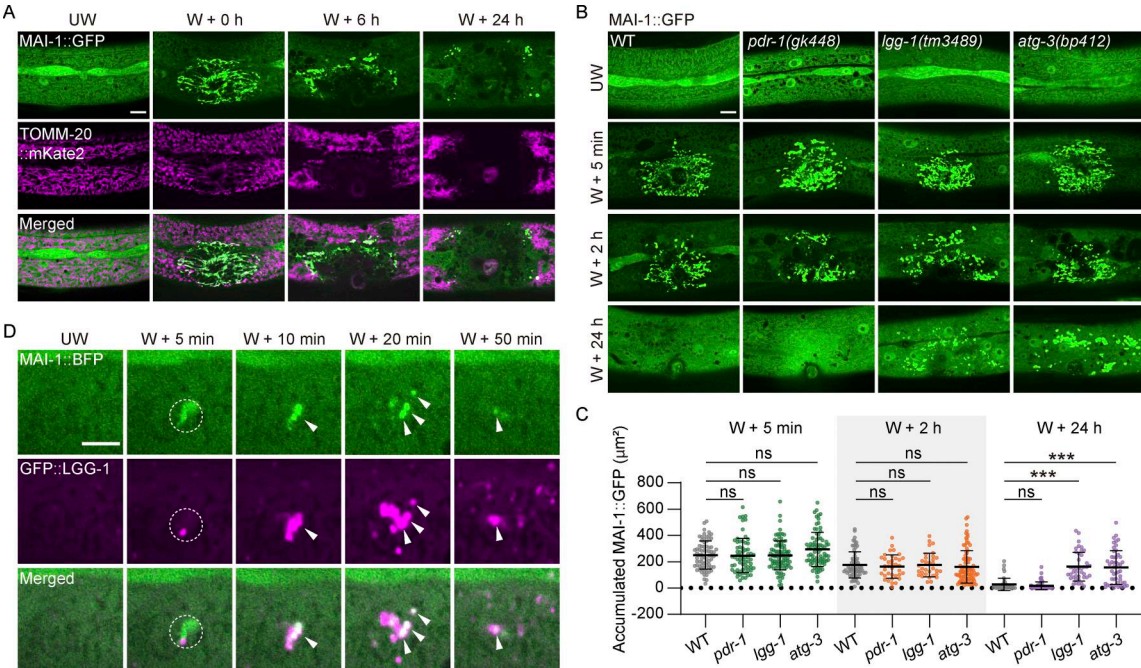

Figure 7. **MAI-1–marked mitochondria undergo autophagy-dependent degradation. (A)** Representative confocal images of MAI-1::GFP and TOMM-20:: mKate2 before and at different time points after laser wounding. Strains: P*hyp-7*-TOMM-20::mKate2(*zjuSi126*); P*col-19*-MAI-1::GFP (*zjuSi510*). Scale bar, 10 µm. **(B)** Representative confocal image of MAI-1::GFP in the epidermal hyp7 after wounding 5 min, 2 h, and 24 h in different genetic backgrounds. Mitophagy related mutants (*pdr-1*, *lgg-1*, and *atg-3*) were used. Strains: *pdr-1(gk448), lgg-1(tm3489), atg-3(bp412)*, and P*col-19*-MAI-1::GFP (*zjuSi510*). Scale bar, 10 µm. **(C)** Quantitation of accumulated MAI-1::GFP in the wounding site in *pdr-1(gk448), lgg-1(tm3489)*, and *atg-3(bp412)* after wounding 5 min, 2 h, and 24 h, shown in B. *n* represents the number of wounds. Statistical analysis was performed using one-way ANOVA. Error bars represent the mean value ± SD (*n* = 72, 67, 91, 78, 65, 36, 32, 92, 44, 64, 41, and 50 from left to right); ***, P < 0.001, compared with WT. **(D)** Representative confocal images of MAI-1::BFP and GFP::LGG-1 at different time points after wounding. The MAI-1 signal is shown in green, while the LGG-1 signal is shown in magenta for better visualization. The white dashed circle indicates the wound site. The white arrow indicates a fusion event. Strains: P*col-19*-GFP::LGG-1(*zjuSi288*); P*col-19*-MAI-1::BFP(*zjuSi539*). Scale bar, 5 µm. (A–D) UW, unwounded; W, wounded.

interacting partners contribute to MAI-1 recruitment remains an important question for future biochemical and structural studies.

The identification of MAI-1 as a damage-specific sensor provides a powerful tool enabling studies not accessible with existing mitochondrial probes. First, MAI-1 allows direct visualization of irreversible mitochondrial membrane damage *in vivo*, distinguishing reversible dysfunction (e.g., depolarization) from terminal structural failure at single-organelle resolution. Second, MAI-1 supports real-time tracking of damaged mitochondria, facilitating analysis of their recognition and clearance by quality control pathways. Third, it enables genetic and chemical screens to identify factors that preserve membrane integrity. Forth, MAI-1 provides a quantitative readout of mitochondrial toxicity induced by physiological or pharmacological stress. Finally, its rapid kinetics make it well suited for studying acute injury paradigms requiring high temporal resolution.

The discovery of MAI-1 as a damage-specific sensor opens new questions about downstream coupling to clearance. In this study, we establish that MAI-1–marked mitochondria are removed through an LGG-1/ATG-3-dependent, PINK1/Parkin-independent autophagy pathway. However, genetic loss-of-function studies will be required to determine whether MAI-1 is an essential receptor or adaptor in this process or whether it

primarily serves as a damage reporter that accompanies, but does not instruct, organelle clearance. Dissecting the *in vivo* function of MAI-1 is further complicated by the presence of a second IF1 homolog in *C. elegans*, MAI-2, which contains a MTS and may have overlapping or compensatory roles. Addressing this question will be essential to determine whether MAI-1 plays a direct functional role in mitochondrial quality control or acts as a parallel damage-sensing module marking membrane damage.

Notably, endogenous MTS-lacking IF1 proteins have not been identified in mammals. However, engineered cIF1 variants, including human IF1, display damage-induced mitochondrial recruitment, suggesting that IF1 family proteins possess an intrinsic capacity to associate with mitochondria when inner membrane components become accessible. Whether such a mechanism operates under physiological conditions in mammalian cells remains unclear. In this context, MAI-1 provides a useful experimental tool for probing mitochondrial membrane integrity and its consequences in higher organisms.

In summary, our work establishes a conceptual framework in which mitochondrial quality control is stratified into functional versus membrane structural surveillance, with MAI-1 representing the first genetically encoded reporter dedicated to membrane damage. Beyond its immediate utility as a tool for visualizing catastrophic mitochondrial damage *in vivo* with ms resolution, MAI-1 opens a new avenue to interrogate the

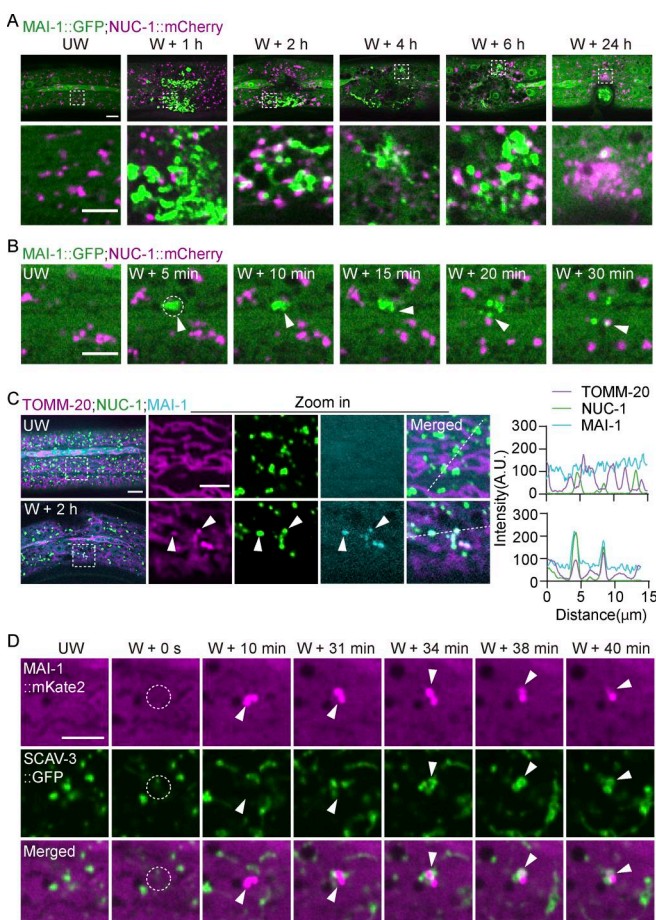

Figure 8. **Lysosomes engulf MAI-1–marked mitochondria after injury.** **(A)** Representative confocal images of MAI-1::GFP and NUC-1::mCherry at different time points after laser wounding. Lysosomes approached and degraded MAI-1–positive mitochondria. Strains: P*ced-1*-NUC-1::mCherry(*qxIs257*); P*col-19*-MAI-1::GFP(*zjuSi510*). Scale bar, 10 and 5 µm (zoom in). **(B)** Time-lapse confocal images of MAI-1::GFP and NUC-1::mCherry after precise single mitochondrial loss. Lysosome fusion coincides with MAI-1 signal loss. The white dashed circle indicates the wound site. The white arrowhead indicates a fusion event. Strains: P*ced-1*-NUC-1::mCherry(*qxIs257*); P*col-19*-MAI-1::GFP(*zjuSi510*). Scale bar, 5 µm. **(C)** Left, representative confocal images of TOMM-20::mYonghong, NUC-1:: mNeonGreen, and MAI-1::BFP before and 2 h after laser-induced wounding. The white arrow indicates colocalized MAI-1–positive mitochondria and lysosome. Right, quantification of fluorescence intensity along 15 µm lines across the wound site. Strains: P*col-19*-NUC-1::mNeonGreen (*zjuSi534*); P*col-19*-MAI-1:: BFP (*zjuSi539*); P*col-19*-TOMM-20::mYonghong (*zjuSi577*). Scale bar, 10 and 5 µm (zoom in). **(D)** Representative time-lapse confocal images of MAI-1:: mKate2 and SCAV-3::GFP before and at different time points after laser wounding. The white arrowhead indicates an engulfing event. Strains: P*scav-3*-SCAV-3::GFP (*qxIs430*); P*col-19*-MAI-1::mKate2(*zjuSi594*). Scale bar, 5 µm. (A–D) UW, unwounded; W, wounded.

mechanism, regulation, and physiological consequences of mitochondrial membrane damage in health and disease.

## Materials and methods

### *C. elegans* and strains
All *C. elegans* strains were maintained at 20–22.5°C on nematode growth medium (NGM) agar plates with *Escherichia coli* OP50.

New strains were constructed using standard procedures, and genotypes were confirmed by PCR and sequencing. Previously described transgenes or mutations include *aass-1 (yq211)*, *argn-1 (yq187)*, *pink-1(ok3538)*, *pdr-1(gk448)*, *fzo-1(tm1133)*, *eat-3(tm1107)*, *drp-1(tm1108)*, *chch-3(tm2336)*, *atg-3(bp412)*, and *lgg-1(tm3489)*. All the strains used are listed in Table S1.

### Constructs and strains
Epidermal MAI-1 was fused to GFP, BFP, mKate2, or mYonghong (Xiong et al., 2025) under the control of the *col-19* promoter (665 bp sequence upstream of start codon). For the endogenous *mai-1* promoter, a 760 bp sequence upstream of the start codon was cloned. GFP, BFP, mKate2, and mYonghong were derived from previously published sources (Meng et al., 2020; Meng et al., 2023; Xiong et al., 2025). All fluorescent proteins were codon-optimized *C. elegans* and contained artificial introns commonly used in transgenes to ensure efficient expression in *C. elegans*. *C. elegans* genomic DNA, cDNA, and human cDNA were used as the template for cloning. Extrachromosomal array transgenic worms were made by injection of constructs at 10 ng/µl with 5 ng/µl co-injection marker (P*myo-2*-RFP). The single-copy insertions P*col-19*-MAI-1::GFP(*zjuSi510*) IV, P*col-19*-MAI-1::BFP(*zjuSi539*) V, and P*col-19*-MAI-1::mKate2 (*zjuSi594*) II were made by the CRISPR-Cas9–based insertion method (Xu et al., 2016). Single-copy insertions were targeted to established genomic loci, including ttTi4348 (Chr I), ttTi5605 (Chr II), cxTi10882 (Chr IV), and ttTi883(Chr V). New transgenic strains are listed in Table S1. All plasmids used in this study are listed in Table S2, and all primers are listed in Table S3.

### Reagents
Reagents, chemicals, and commercial kits used in this study are listed in Table S4.

### Needle and laser wound
Worms were wounded with single stabs of a microinjection needle to either the anterior or posterior body 24 h after the L4 stage, essentially as described elsewhere (Wijaya et al., 2020). Laser wound: Worms were wounded with a Micropoint UV laser to the anterior or posterior body 24 h after the L4 stage. The energy ranged from 65 to 70. The repetition rate was 10 Hz, which repeats three times.

### Ultrasound wound
Approximately 200 adult worms were transferred into 200 µl of M9 buffer containing *E. coli* OP50. The sonicator probe tip (JU92-IIN; Ultrasonic homogenizer, Scientz) was immersed below the liquid surface. Sonication was performed at 20% amplitude using a pulsed regimen of 0.1-s working intervals and 3-s resting intervals. Worms were subjected to four cycles of sonication.

### Cold stress
NGM plates containing adult worms were placed at –20°C for 3 h. Subsequently, the plates were allowed to thaw to room temperature. After the plates had thawed, worms were transferred and prepared on microscope slides. Freezing or subzero stress induces mechanical membrane damage, osmotic stress, and ROS

production that can physically compromise mitochondrial membranes (Moussa et al., 2008).

## Optogenetic-induced damage
miniSOG(Q103L) was fused to the N-terminal 55 amino acids of the OMM protein TOMM-20 as previously described (Xu and Chisholm, 2016; Qi et al., 2012). The expression was driven in the adult epidermis using the *col-19* promoter. 20–30 young adults were placed centrally on 3-cm diameter NGM plates with 100 μl of 100 mM $CuCl_2$ applied around the periphery of the agar surface. The entire NGM plate was uniformly illuminated with blue light (Arbitrary function signal generator, FY3200S; FeelTech).

## Drug treatment
Drugs were added to NGM plates and allowed to dry for 1–2 h at room temperature before transferring the young adult worms. FCCP (C2920; Sigma-Aldrich) was dissolved in DMSO (D4540; Sigma-Aldrich) as a 10 mM stock solution, and its final concentration was 15 μM. Rotenone (R8875; Sigma-Aldrich) was dissolved in DMSO as a 40 mM stock solution, and its final concentration was 100 μM. Oligomycin (T21494; TargetMol) was dissolved in DMSO as a 1 mM stock solution, and its final concentration was 10 μM. The synchronized young adults were transferred to a freshly made NGM drug plate for 2 h and then imaged using the spinning disk confocal.

## TMRE staining
TMRE (T669; Thermo Fisher Scientific) was added to the seeded NGM from a 10 mM stock to a final concentration of 5 μM and dried for 1–2 h at room temperature. Young adult worms were transferred to the plates for 2 h and imaged.

## RNAi knockdown in worms
RNAi was performed as previously described (Meng et al., 2020). Briefly, synchronized L4 stage worms were placed on HT115 *E. coli* seeded NGM containing 1 mM IPTG and carbenicillin and incubated at 20°C overnight before imaging.

## Confocal imaging
Images were acquired using a spinning-disk confocal microscope Nikon Eclipse Ti equipped with an Andor confocal scanning unit and a 100 × /1.46 NA oil immersion objective (Nikon). Imaging was performed at room temperature ranging from 20 to 25°C. To stabilize the positions of the epidermis, worms were immobilized with 12 mM levamisole and mounted on 4% agarose. Z-section and time-lapse were controlled using Andor IQ software. The EMCCD camera gain was set to 100 for all channels. The imaging region of interest (ROI) is 1024 × 1024 pixels. Fluorescence was excited using 488 nm (GFP/mNeonGreen), 561 nm (RFP/mKate2/mCherry/mYonghong/TMRE), and 405 nm (BFP) lasers. Under basal conditions, the exposure time was 200 ms for the GFP and RFP channels and 400 ms for the BFP channel. For laser wounding experiments, images were acquired as single confocal planes, with one frame captured before wounding. For ms-scale time-lapse imaging, the GFP channel was imaged with a 100-ms exposure time at a 200-ms acquisition interval. Detailed acquisition parameters for each experiment are provided in the corresponding video legends. All images were analyzed using Fiji (ImageJ, https://imagej.net/software/fiji/downloads).

## High intelligent and sensitive SIM imaging
Super resolution imaging was performed using a commercial High Intelligent and Sensitive SIM (HIS-SIM) system (CSR Biotech), following the previously described procedures (Huang et al., 2018). Images were captured using a 100 × /1.5 NA oil immersion objective (Olympus) under control of the manufacturer's software. Imaging was performed at room temperature ranging from 20 to 25°C. For laser wounding experiments, mitochondria were wounded using a Micropoint UV laser on a spinning-disk confocal microscope and then transferred to the HIS-SIM system. Exposure times were set to 10 ms for the GFP and RFP channels and 100 ms for the BFP channel. The imaging ROI is described in the corresponding figure legends. Raw SIM datasets were reconstructed using a Wiener filter–based algorithm for defocus elimination, artifact removal, and image denoising. To further enhance the resolution and contrast in fluorescence images, sparse deconvolution was applied as previously described (Zhao et al., 2022).

## Image analysis
Three regions of the unwounded/wounded/background region were selected in the same images, and their average intensities were calculated to measure the intensity of the unwounded/wounded/background region $(I_{uw}/I_w/I_{bg})$. $(I_w-I_{uw})/(I_{uw}-I_{bg})$ was calculated to obtain the fluorescence signal (ΔFw/Fuw) at the wounded region.

## Quantification of accumulated MAI-1::GFP Area
Z-stack images were acquired and subjected to Z-projection using maximum intensity projection. The background signal was minimized by adjusting the threshold levels. ROIs were manually defined using the Freehand selection tool in Fiji. The total area of accumulated GFP within the selected ROIs was quantified using the Analyze Particles command in Fiji as previously described (Hu et al., 2023).

## Colocalization analysis
The colocalization analysis was conducted by calculating Pearson's correlation coefficient (PCC) as described (Bolte and Cordelières, 2006). The PCC value ranges from –1 to 1, with 1 indicating complete colocalization and –1 indicating perfect negative correlation, with zero indicating no correlation. For analysis, the ROIs (6.9 × 6.9 μm) were selected in the captured images. The region at the wound site was selected for wounded worms as the ROI. The quantitative value of PCC was calculated in ImageJ by the following options: Analyze-Colocalization-Coloc 2.

## Quantitative analysis
All experiments were conducted a minimum of three times to ensure accuracy and reproducibility. Statistical analyses were performed using GraphPad Prism 9. The mean of the individual

data was plotted on the Y-axis of the graph, with the SD of the mean shown as an error bar. Two-group comparisons were performed using the Mann–Whitney test. Multiple-group comparisons were analyzed with the one-way ANOVA. Statistically significant was defined as $P < 0.05$ and is indicated with asterisks (*$P < 0.05$, **$P < 0.01$, and ***$P < 0.001$). Data distribution was assumed to be normal, but this was not formally tested. At least 10 different wound sites were quantified in each experiment, and each experiment was repeated at least twice.

### Phylogenetic analysis
Protein sequences of MAI-1 homologs from representative eukaryotic species were retrieved from UniProt (Consortium et al., 2024), NCBI (Sayers et al., 2024), and WormBase (Sternberg et al., 2024). Multiple sequence alignments were generated using MAFFT (v7.505) with default parameters (Katoh and Standley, 2013). Phylogenetic trees were constructed using IQ-TREE (v3.0.1) (Minh et al., 2020) with model selection enabled (-m MFP), bootstrap replicates (-bb 1000), nearest neighbor interchange (-bnni), automatic CPU detection (-nt AUTO), and a maximum of 15 categories of FreeRate model (-cmax 15). MTSs were predicted using TargetP (v2.0) (Armenteros et al., 2019). Phylogenetic tree and predicted mitochondrial localization annotations were visualized using *ggtree* and *ggtreeExtra* package in R (v4.5) (Yu, 2020).

### Protein identity matrix
Pairwise sequence alignment of *C. elegans* MAI-1, MAI-2, and human ATP5IF1 was performed using UniProt online alignment tools (Consortium et al., 2024). Predicted identity values were extracted and visualized as a heatmap using the *pheatmap* package in R (v4.5) (Yu, 2020).

### Online supplemental material
Fig. S1 shows that laser wounding triggers mitochondrial membrane damage and clearance. Fig. S2 shows that the MAI-1 marks damaged mitochondria after laser wounding. Fig. S3 shows that the MAI-1 labels damaged mitochondria induced by diverse stresses. Fig. S4 shows that mitochondrial depolarization is insufficient to trigger MAI-1 translocation. Fig. S5 shows that MAI-1–marked mitochondria are subject to lysosomal degradation. Table S1 lists the strains used in this study. Table S2 lists the plasmids used in this study. Table S3 lists the primers used in this study. Table S4 lists the reagents, chemicals, and commercial kits used in this study. Video 1 shows that epidermal wounding induces mitochondrial fragmentation and a rapid reduction in mitochondrial membrane potential. Video 2 shows outer and inner membrane damage accompanied by rapid loss of matrix signal and membrane potential. Video 3 shows that epidermal wounding induces MAI-1 relocalization. Video 4 shows that epidermal wounding induces MAI-1 translocation to mitochondria. Video 5 shows that epidermal wounding triggers the translocation of cytosolic MAI-2 and IF1 to damaged mitochondria. Videos 6 and 7 show that MAI-1 translocates to mitochondria with sustained, but not transient, membrane potential loss. Video 8 shows that epidermal wounding induces fragmentation of MAI-1–marked mitochondria and their fusion with LGG-1. Video 9 shows that

epidermal wounding induces fragmentation of MAI-1–marked mitochondria and their fusion with the lysosomes. Video 10 shows that epidermal wounding induces fragmentation of MAI-1–marked mitochondria and their enclosure by the lysosomes.

## Data availability
The data are available from the corresponding author upon reasonable request.

## Acknowledgments
We thank Drs. Haoxing Xu, Hong Zhang, Xiaochen Wang, and Junjie Hu for their comments and valuable discussion. We thank the Imaging Center of Zhejiang University School of Medicine for assistance with confocal microscopy. We thank Dr. Guifeng Xiao from the Core facilities of Zhejiang University School of Medicine for imaging support, analysis, and reconstruction.

This work was supported by the National Key R&D Program of China (2021YFA1300302 and 2021YFA1101002), the National Natural Science Foundation of China (92254303), and the Zhejiang Provincial Natural Science Foundation of China under grant no. LRG26C070001 to S. Xu. Open Access funding provided by University of Edinburgh.

Author contributions: Erqing Gao: conceptualization, data curation, formal analysis, investigation, methodology, validation, visualization, and writing—original draft, review, and editing. Liwei Guan: investigation. Kehang Zhang: data curation, formal analysis, funding acquisition, investigation, methodology, resources, software, visualization, and writing—original draft, review, and editing. Shuzhe Qi: visualization. Jingxiu Xu: investigation. Jie Zhang: investigation. Tianyi Zhu: investigation. Bing Yang: investigation. Chonglin Yang: resources and supervision. Eric Miska: conceptualization, supervision, and writing—review and editing. Mao Zhang: conceptualization, methodology, resources, validation, and writing—review and editing. Suhong Xu: conceptualization, data curation, formal analysis, funding acquisition, investigation, methodology, project administration, resources, supervision, visualization, and writing—original draft, review, and editing.

Disclosures: The authors declare no competing interests exist.

Submitted: 12 November 2025

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

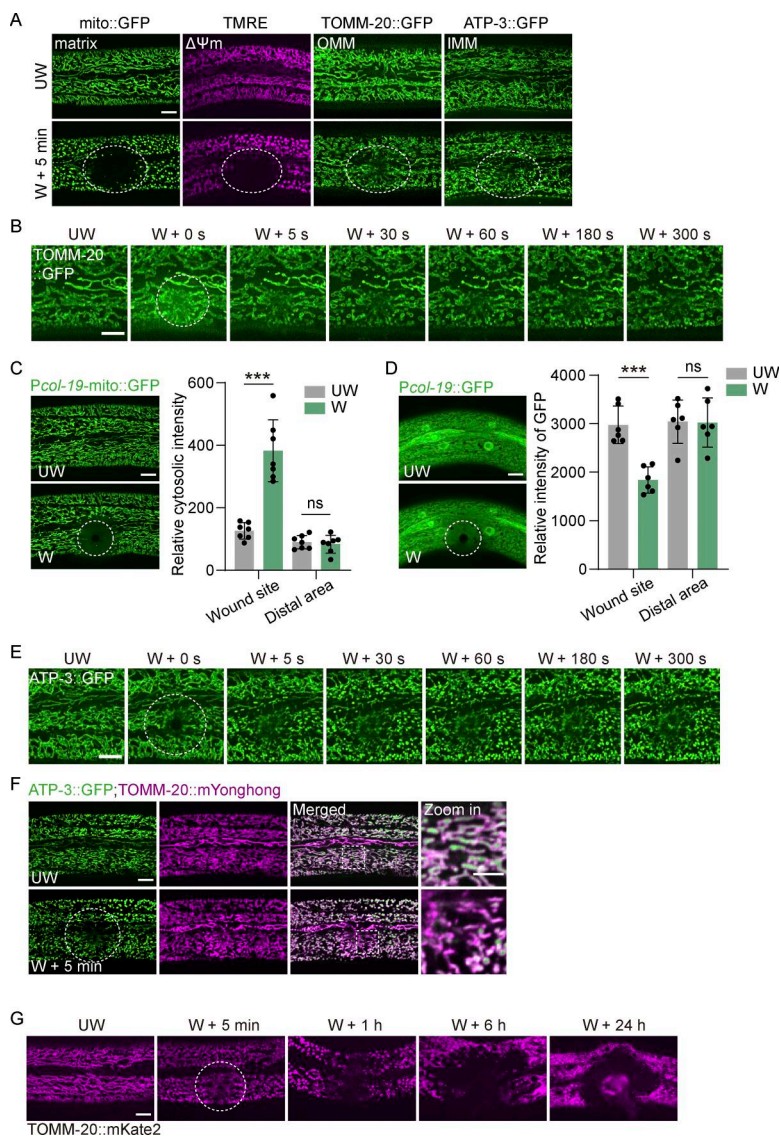

**Figure S1. Laser wounding triggers mitochondrial membrane damage and clearances. (A)** Representative confocal image showing mito::GFP, TMRE, TOMM-20::GFP, and ATP-3::GFP before and after wounding. Wounding triggers loss of mitochondrial matrix and membrane potential (ΔΨm). P*col-19*-mito::GFP (*zjuSi197*), P*col-19*-TOMM-20::GFP (*zjuSi42*), and ATP-3::GFP (KI, *zju133*) animals were used for imaging and wounding. Scale bar, 10 μm. **(B)** Representative time-lapse confocal images of TOMM-20::GFP before and after laser wounding. Strain: P*col-19*-TOMM-20::GFP(*zjuSi42*). Scale bar, 10 μm. **(C)** Left, representative confocal image of mito::GFP before and 50 ms after wounding. Strains: P*col-19*-mito::GFP (*zjuSi197*). Scale bar, 10 μm. Right, quantification of relative cytosolic fluorescence intensity before and after wounding. Fluorescence was measured from ten cytosolic ROIs (0.476 μm × 0.476 μm) surrounding mitochondria at the wound site and at distal regions per animal. Statistical analysis was performed using the Mann–Whitney test. Data are mean ± SD (*n* = 7 animals per group); ***, P < 0.001. **(D)** Left, representative confocal image showing diffuse GFP before and 200 ms after wounding. Strains: P*col-19*::GFP (*zjuSi60*). Scale bar, 10 μm. Right, quantification of relative GFP intensity before and after wounding. Fluorescence was measured from 10 cytosolic ROI (0.476 μm × 0.476 μm) surrounding mitochondria at the wound site and at distal regions per animal. Statistical analysis was performed using the Mann–Whitney test. Data are mean ± SD (*n* = 6 animals per group); ***, P < 0.001. **(E)** Representative time-lapse confocal images of ATP-3::GFP before and after laser wounding. Strain: ATP-3::GFP (KI, *zju133*). Scale bar, 10 μm. **(F)** Representative spinning-disk confocal images of ATP-3::GFP and TOMM-20::mYonghong before and after laser wounding. Strains: *ATP-3::GFP*(*zju133*); P*col-19*-TOMM-20::mYonghong (*zjuSi577*). Scale bar, 10 and 5 μm (zoom in). **(G)** Representative confocal images of TOMM-20::mKate before and at different time points after laser wounding. Strain: P*hyp-7*-TOMM-20::mKate2 (*zjuSi126*). Scale bar, 10 μm. **(A–G)** The white dashed circle indicates the wound site. UW, unwounded; W, wounded.

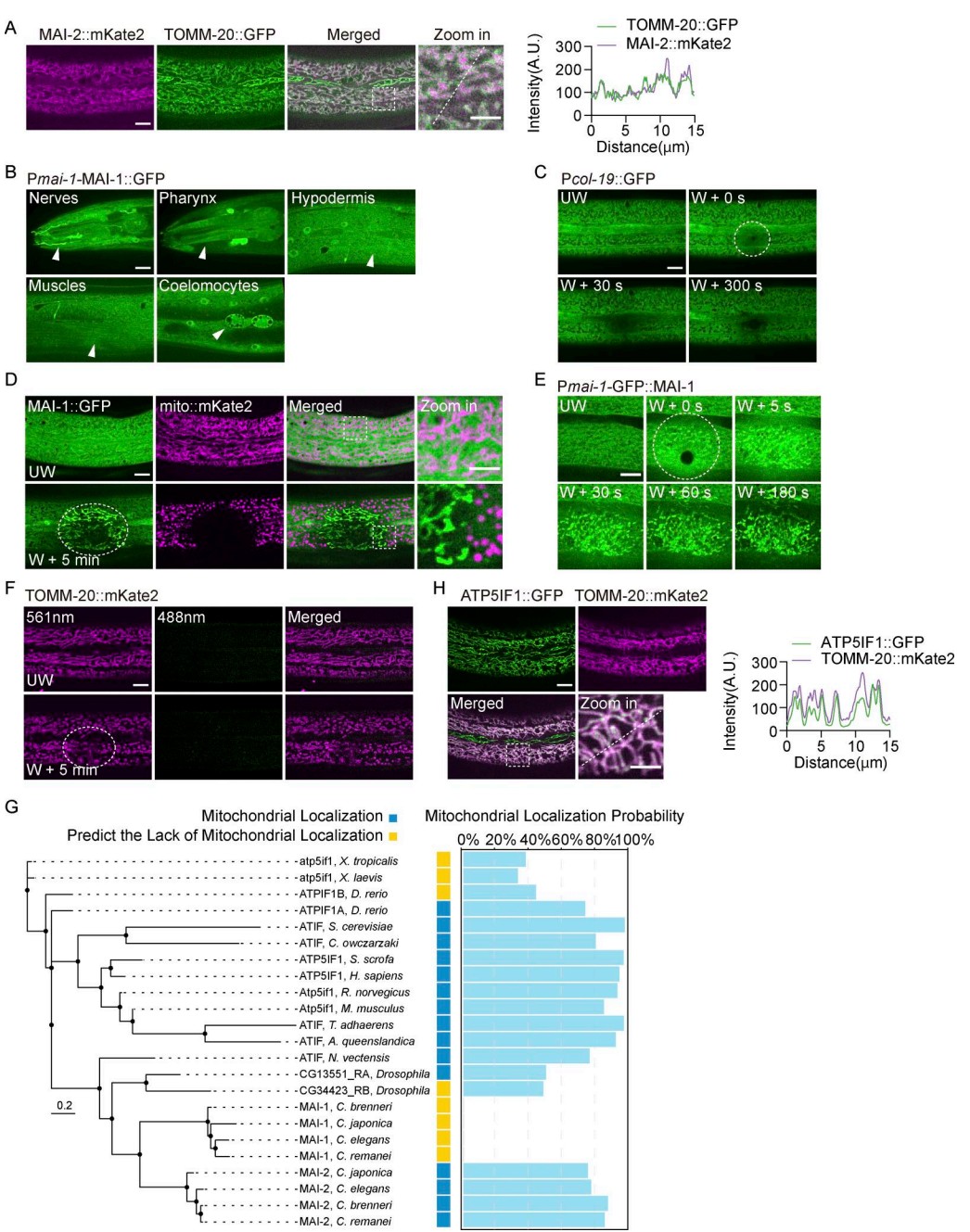

Figure S2. **MAI-1 marks damaged mitochondria after laser wounding. (A)** Left, representative confocal images of MAI-2::mKate2 and TOMM-20::GFP under basal conditions. Strains: Pcol-19-TOMM-20::GFP (zjuSi42);Pcol-19-MAI-2::mKate2 (zjuEx3210). Scale bar, 10 and 5 µm (zoom in). Right, quantification of fluorescence intensity along 15 µm lines across the images on the left. The white dashed lines indicate the analysis region. **(B)** Representative confocal images of Pmai-1::MAI-1::GFP expression in different tissues. Strains: Pmai-1-MAI-1::GFP (zjuSi602). Scale bar: 10 µm. The white arrowhead indicates the expression tissue. **(C)** Representative time-lapse confocal images of Pcol-19::GFP before and after laser wounding. Strains: Pcol-19::GFP (zjuSi60). Scale bar: 10 µm. The white dashed circle indicates the wound site. UW, unwounded; W, wounded. **(D)** Representative confocal images of MAI-1::GFP; mito::mKate2 before and after laser wounding. Strains: Pcol-19-mito::mKate2 (zjuSi47); Pcol-19-MAI-1::GFP (zjuEx2864). Scale bar: 10 and 5 µm (zoom in). The white dashed circle indicates the wound site. UW, unwounded; W, wounded. **(E)** Representative time-lapse confocal images of GFP::MAI-1 before and after laser wounding. Strain: Pmai-1-GFP:: MAI-1 (zjuEx3038). Scale bar: 10 µm. UW, unwounded; W, wounded. **(F)** Representative confocal images of TOMM-20::mKate2 before and after laser wounding using 561 and 488 nm emission channel. Strains: Phyp-7-TOMM-20::mKate2 (zjuSi126). Scale bar, 10 µm. The white dashed circle indicates the wound site. UW, unwounded; W, wounded. **(G)** Phylogenetic analysis and predicted mitochondrial localization of ATP5IF1 homologs. Left, phylogenetic tree of ATP5IF1 homologs from multiple eukaryotic species, including nematodes, mammals, and early metazoans. Right, predicted probability of containing mitochondrial localization sequences. Proteins with a prediction probability of > 50% are considered to have a mitochondrial localization signal. **(H)** Left, representative confocal images of ATP5IF1::GFP and TOMM-20::mKate2 under basal conditions. Strains: Pcol-19-ATP5IF1::GFP (zjuEx3485);Phyp-7-TOMM-20::mKate2 (zjuSi126). Scale bar, 10 and 5 µm (zoom in). Right, quantification of fluorescence intensity along 15 µm lines across the images on the left. The white dashed lines indicate the analysis region.

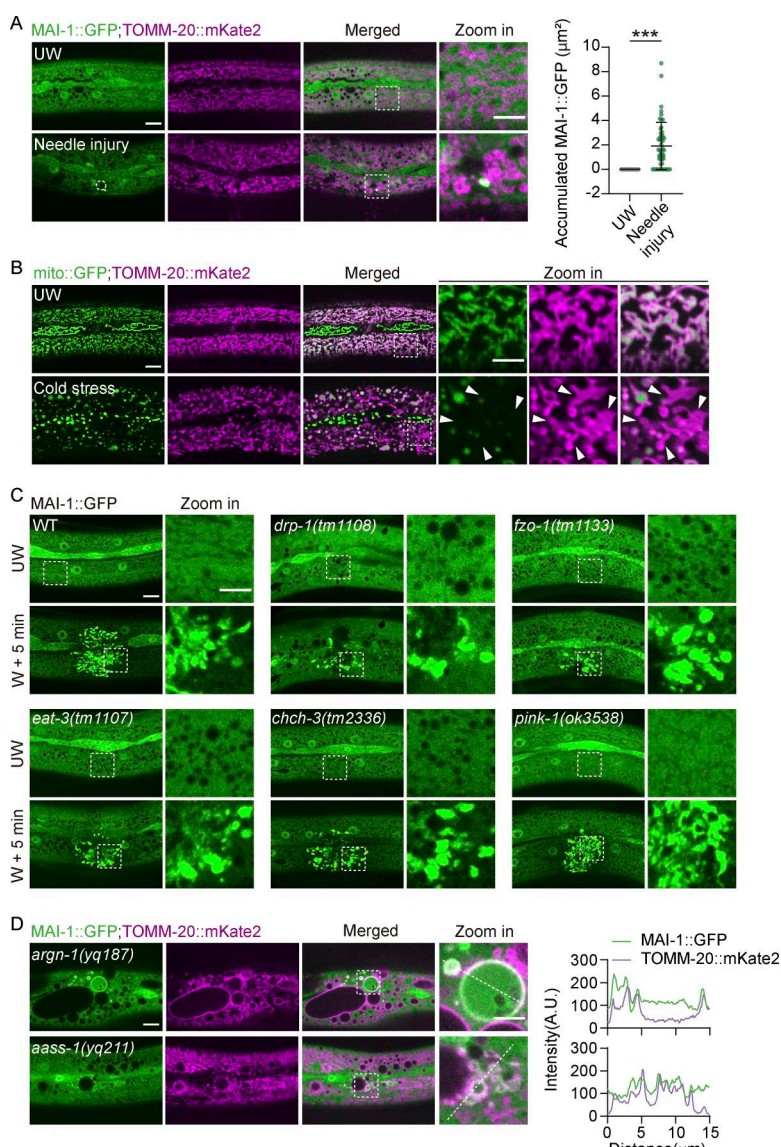

Figure S3. **MAI-1 labels damaged mitochondria induced by diverse stresses. (A)** Left, representative confocal image of MAI-1::GFP and TOMM-20::mKate2 in the epidermal hyp7 before and after mechanical puncture. Strains: P*hyp-7*-TOMM-20::mKate2 (*zjuSi126*); P*col-19*-MAI-1::GFP (*zjuSi510*). The white dashed circle indicates the wound site. Scale bar, 10 and 5 μm (zoom in). UW, unwounded. Right, quantitation of accumulated MAI-1::GFP in the wounding site before and after wounding 5 min. Statistical analysis was performed using the Mann–Whitney test. Data are mean ± SD (*n* = 39 and 52, respectively). ***P < 0.001. **(B)** Representative confocal image of mito::GFP and TOMM-20::mKate2 in the epidermal hyp7 before and after cold shock (−20°C) stress. The white arrowheads indicate damaged mitochondria. Strains: P*hyp-7*-TOMM-20::mKate2 (*zjuSi126*); P*col-19*-mito::GFP (*zjuSi197*). Scale bar, 10 and 5 μm (zoom in). **(C)** Representative confocal image of MAI-1::GFP in the epidermal hyp7 before and after wounding in different genetic backgrounds. Strains: *fzo-1*(*tm1133*), *drp-1*(*tm1108*), *eat-3*(*tm1107*), *chch-3*(*tm2336*), *pink-1*(*ok3538*), and P*col-19*-MAI-1::GFP (*zjuSi510*). Scale bar, 10 and 5 μm (zoom in). UW, unwounded; W, wounded. **(D)** Representative confocal image of MAI-1::GFP and TOMM-20::mKate2 in *aass-1* and *argn-1* mutant. Enlarged and damaged mitochondria colocalize with MAI-1::GFP. Strains: *aass-1* (*yq211*), *argn-1* (*yq187*); P*hyp-7*-TOMM-20::mKate2 (*zjuSi126*); P*col-19*-MAI-1::GFP (*zjuSi510*). Scale bar, 10 and 5 μm (zoom in). Right, quantification of fluorescence intensity along 15 μm lines across the wound site. The white dashed lines indicate the analysis region.

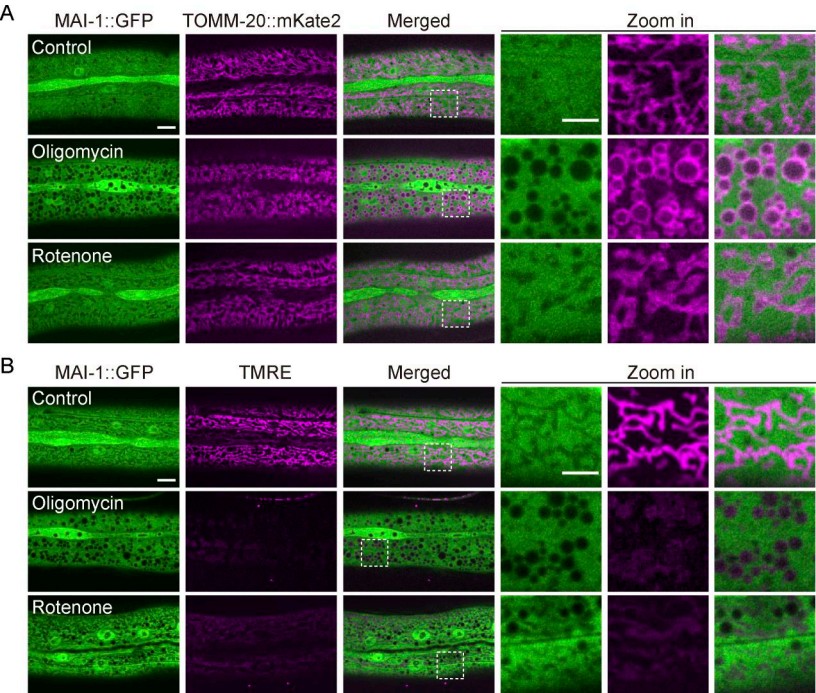

Figure S4. **Mitochondrial depolarization is insufficient to trigger MAI-1 translocation. (A)** Representative confocal images of MAI-1::GFP and TOMM-20::mKate2 after oligomycin (10 µM, 2 h) or rotenone (100 µM, 2 h). Strain: P*hyp-7*-TOMM-20::mKate2 (*zjuSi126*); P*col-19*-MAI-1::GFP *(zjuSi510)*. Scale bar, 10 and 5 µm (zoom in). **(B)** Representative confocal images of MAI-1::GFP and TMRE after oligomycin (10 µM, 2 h) or rotenone (100 µM, 2 h). TMRE was diminished, but MAI-1 remained cytosolic after 2 h of oligomycin and rotenone treatment. Strains: P*col-19*-MAI-1::GFP (*zjuSi510*). Scale bar: 10 and 5 µm (zoom in).

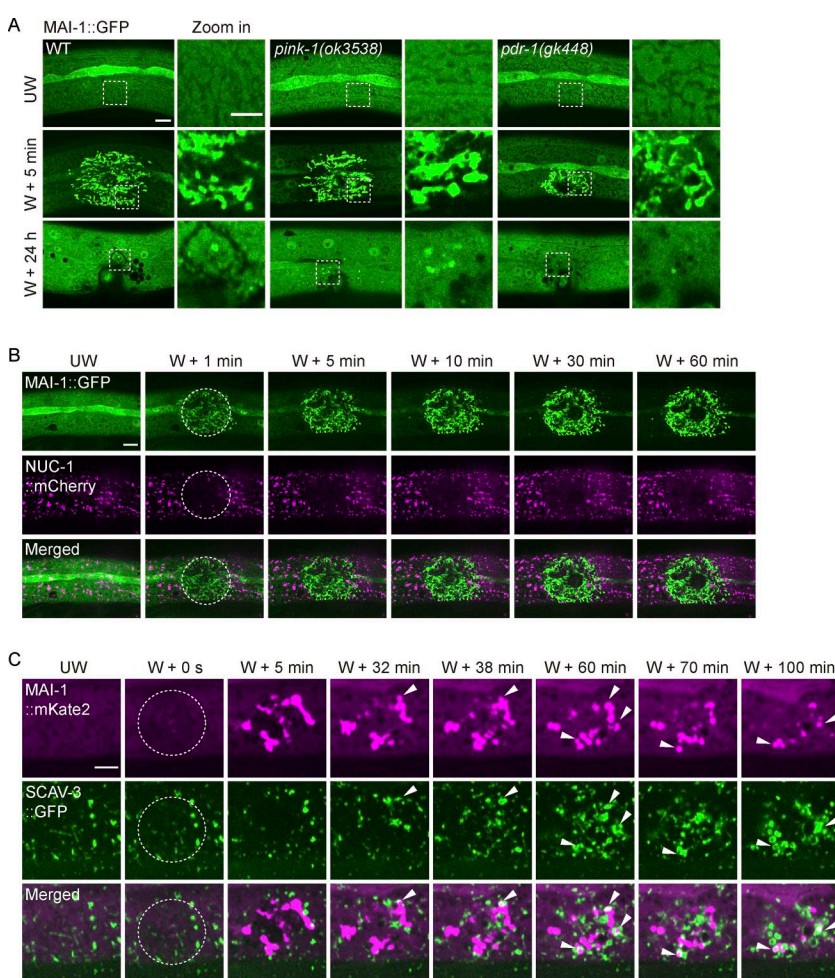

Figure S5.    **MAI-1–marked mitochondria are subject to lysosomal degradation. (A)** Representative confocal images of MAI-1::GFP at 5 min and 24 h after wounding in *pink-1* and *pdr-1* mutant backgrounds. Clearance proceeded normally. Strains: *pink-1(ok3538)*, *pdr-1(gk448)*, and P*col-19*-MAI-1::GFP (*zjuSi510*). Scale bar, 10 and 5 µm (zoom in). **(B)** Representative time-lapse confocal images of MAI-1::GFP and NUC-1::mCherry during 1–60 min after laser wounding. Strain: P*ced-1*-NUC-1::mCherry (*qxIs257*); P*col-19*-MAI-1::GFP (*zjuSi510*). Scale bar, 10 µm. The white dashed circle indicates the wound site. **(C)** Representative time-lapse confocal images of MAI-1::mKate2 and SCAV-3::GFP before and at different time points after laser wounding. Strains: P*scav-3*-SCAV-3::GFP (*qxIs430*); P*col-19*-MAI-1::mKate2 (*zjuSi594*). Scale bar, 5 µm. The white dashed circle indicates the wound site. The white arrowhead indicates enveloping event. **(A–C)**: UW, unwounded; W, wounded.

Video 1.    **Epidermal wounding induces mitochondrial fragmentation and a rapid reduction in mitochondrial membrane potential.** Strain: P*col-19*-mito::GFP (*zjuSi197*). Frames were collected at 1 frame per second (fps) and displayed at 30 fps. Scale bar, 10 µm. Related to Fig. 1 A.

Video 2.    **Outer and inner membrane damage accompanied by rapid loss of matrix signal and membrane potential.** Left, epidermal wounding induces mitochondrial fragmentation, outer membrane damage, and immediate dissipation of matrix signal. Middle, inner membrane damage with immediate reduction in mitochondrial membrane potential. Right, combined outer and inner membrane damage. Strain: P*hyp-7*-TOMM-20::mKate2 (*zjuSi126*); P*col-19*-mito::GFP (*zjuSi197*); P*col-19*-TOMM-20::mYonghong (*zjuSi577*); ATP-3::GFP (KI, *zju133*). Scale bar, 10 µm. Frames were collected at 5 fps and displayed at 10 fps. Related to Fig. 1, B and C; and Fig. S1 F. fps, frame per second.

Video 3.    **Epidermal wounding induces MAI-1 relocalization. Strain: P*col-19*-MAI-1::GFP(*zjuSi510*).** Scale bar, 10 µm. Frames were collected at 5 fps and displayed at 60 fps. Related to Fig. 2 B. fps, frame per second.

Video 4.    **Epidermal wounding induces MAI-1 translocation to mitochondria.** Strain: P*col-19*-MAI-1::GFP (*zjuSi510*);P*hyp-7*-TOMM-20::mKate2 (*zjuSi126*). Scale bar, 10 µm. Frames were collected at 5 fps and displayed at 10 fps. Related to Fig. 2 E. fps, frame per second.

Video 5.    **Left, epidermal wounding induces cytosolic MAI-2 translocation to the mitochondria.** Right, epidermal wounding induces cIF1 translocation to the mitochondria. Magenta represents TOMM-20 and green represents cMAI-2 and cIF1, respectively. Strain: P*hyp-7*-TOMM-20::mKate2 (*zjuSi126*); P*col-19*-IF1(ΔMTS)(20–106 aa)::GFP (*zjuEx3447*);P*col-19*-MAI-2(ΔMTS)(18–109 aa)::GFP (*zjuEx3449*). Frames were collected at 5 fps and displayed at 30 fps. Scale bar, 10 µm. Related to Fig. 3, C and D. fps, frame per second.

Video 6.   **MAI-1 translocates to mitochondria with sustained, but not transient, membrane potential loss.** The yellow arrowhead (a1) indicates irreversibly depolarized mitochondria. The white arrowhead (a2) indicates recovered mitochondria. Frames were collected at 5 fps and displayed at 5 fps. Strain: P*col-19*-MAI-1::GFP(*zjuSi510*). Scale bar, 10 μm. Related to Fig. 5 A. fps, frame per second.

Video 7.   **MAI-1 translocates to mitochondria with sustained, but not transient, membrane potential loss.** Strain: P*col-19*-MAI-1::GFP(*zjuSi510*). The yellow arrowhead indicates irreversibly depolarized mitochondria, and the white arrowhead indicates recovered mitochondria. Frames were collected at 2 fps and displayed at 10 fps. Scale bar, 10 μm. Related to Fig. 5 D. fps, frame per second.

Video 8.   **Epidermal wounding induces fragmentation of MAI-1–marked mitochondria and their fusion with LGG-1.** Green represents MAI-1, and magenta represents LGG-1. The white arrowhead indicates a fusion event. Strains: P*col-19*-GFP::LGG-1 (*zjuSi288*); P*col-19*-MAI-1::BFP (*zjuSi539*). Frames were collected at 10 fps and displayed at 20 fps. Scale bar, 5 μm. Related to Fig. 7 D. fps, frame per second.

Video 9.   **Epidermal wounding induces fragmentation of MAI-1–marked mitochondria and their fusion with the lysosomes.** Green represents MAI-1, and magenta represents NUC-1. The white arrowhead indicates the fusion event. Strains: P*ced-1*-NUC-1::mCherry (*qxIs257*); P*col-19*-MAI-1::GFP (*zjuSi510*). Frames were collected at 2 fps and displayed at 30 fps. Scale bar, 5 μm. Related to Fig. 8 B. fps, frame per second.

Video 10.   **Epidermal wounding induces fragmentation of MAI-1–marked mitochondria and their enclosure by the lysosomes.** The white arrowhead indicates the enveloping event. Strains: P*scav-3*-SCAV-3::GFP (*qxIs430*); P*col-19*-MAI-1::mKate2 (*zjuSi594*). Frames were collected at 10 fps and displayed at 20 fps. Scale bar, 5 μm. Related to Fig. 8 D. fps, frame per second.

**Provided online are Table S1, Table S2, Table S3, and Table S4. Table S1 lists the strains used in this study. Table S2 lists the plasmids used in this study. Table S3 lists the primers used in this study. Table S4 lists the reagents, chemicals, and commercial kits used in this study.**

