## [Peer Review File · The Journal of Cell Biology]

A cytosolic IF1 reporter enables real-time visualization of severe mitochondrial membrane damage

Erqing Gao, Liwei Guan, Kehang Zhang, Shuzhe Qi, Jingxiu Xu, Jie Zhang, Tianyi Zhu, Bing Yang, Chonglin Yang, Eric Miska, Mao Zhang, and Suhong Xu

Corresponding Author(s): Suhong Xu, Zhejiang University

Review Timeline:

Submission Date:	2025-11-12
Editorial Decision:	2025-12-10
Revision Received:	2026-02-13
Editorial Decision:	2026-03-10
Revision Received:	2026-04-10
Editorial Decision:	2026-04-23
Revision Received:	2026-05-03
Accepted:	2026-05-04

Monitoring Editor: Hong Zhang

Scientific Editor: Andrea Marat

Transaction Report:

DOI: <https://doi.org/10.1083/jcb.202511088>

December 10, 2025

Re: JCB manuscript #202511088

Suhong Xu
Zhejiang University

Dear Prof. Xu,

Thank you for submitting your manuscript entitled "A cytosolic IF1 senses mitochondrial membrane rupture". The manuscript was assessed by expert reviewers, whose comments are appended to this letter. We invite you to submit a revision if you can address the reviewers' key concerns, as outlined here.

You will see that the reviewers appreciate the potential importance for the field of a genetically encoded reporter for mitochondrial membrane rupture. We agree that a suitably revised study is appropriate for publication in JCB under our Tool format. As such, all reviewer comments pertaining to better characterizing and validating MAI-1 as a reporter must be completely addressed, along with all issues regarding presentation and clarity. However, for a Tool we do not expect you to examine the physiological roles of MAI-1 in elimination of damaged mitochondria as providing detailed insight into the biological significance of findings is outside of the requirement for JCB Tools. We look forward to future work examining this question in detail.

GENERAL GUIDELINES:

Text limits: Character count for an Report is < 40,000, not including spaces. Count includes title page, abstract, introduction, results, discussion, and acknowledgments. Count does not include materials and methods, figure legends, references, tables, or supplemental legends.

Figures: Reports may have up to 10 main text figures. Figures must be prepared according to the policies outlined in our Instructions to Authors, under Data Presentation, <https://jcb.rupress.org/site/misc/ifora.xhtml>. All figures in accepted manuscripts will be screened prior to publication.

Supplemental information: There are strict limits on the allowable amount of supplemental data. Reports may have up to 5 supplemental figures. Up to 10 supplemental videos or flash animations are allowed. A summary of all supplemental material should appear at the end of the Materials and methods section.

Please note that JCB now requires authors to submit Source Data used to generate figures containing gels and Western blots with all revised manuscripts. This Source Data consists of fully uncropped and unprocessed images for each gel/blot displayed in the main and supplemental figures. For assays performed using capillary electrophoresis and/or immunoassay-based detection, authors should instead provide the electropherogram graph(s) for each experiment, plotting fluorescence/chemiluminescence intensity vs. molecular weight/size. Please be sure to provide one Source Data file for each figure gels, blots, and/or capillary electrophoresis assays along with your revised manuscript files. File names for Source Data figures should be alphanumeric without any spaces or special characters (i.e., SourceDataF#, where F# refers to the associated main figure number or SourceDataFS# for those associated with Supplementary figures). For traditional gels and blots, the lanes of the gels/blots should be labeled as they are in the associated figure, the place where cropping was applied should be marked (with a box), and molecular weight/size standards should be labeled wherever possible. For capillary electrophoresis assays, each trace in the graph should be color-coded and labeled to indicate which protein, gene, or sample is being measured (please try to avoid red/green combinations to accommodate our color-blind readers).

The typical timeframe for revisions is three to four months. If you anticipate any difficulties in meeting this aforementioned

revision time limit, please contact us and we can work with you to find an appropriate time frame for resubmission. Please note that papers are generally considered through only one revision cycle, so any revised manuscript will likely be either accepted or rejected.

Thank you for this interesting contribution to Journal of Cell Biology. You can contact us at the journal office with any questions at cellbio@rockefeller.edu.

Sincerely,

Hong Zhang, PhD
Monitoring Editor

Andrea L. Marat, PhD
Deputy Editor

Journal of Cell Biology

Reviewer #1 (Comments to the Authors (Required)):

The study by Gao et al identifies IF1 as a sensor of mitochondrial membrane rupture. They show that the cytosolic IF1 in *C. elegans* localizes to mitochondria after laser damage and some other types of mitochondrial damage. This property can be engineered with human IF1 by removing the mitochondrial targeting sequence. They suggest that this provides a new tool to study mitochondrial physiology.

Overall, the conclusion of the study is interesting, because it indeed would be useful to have a genetically encoded reporter of mitochondrial membrane integrity. However, the conclusions made by the authors need more supporting evidence to be fully convincing.

The study relies entirely on a single technique--the imaging of fluorescent reporters after a perturbation. This limits their ability to interpret the physical basis of their observation, and some of their statements appear overinterpreted. Second paragraph of the Results says that "After wounding, TOMM-20::mKate2 became discontinuous surrounding the ATP-3::GFP (Figure 1B), consistent with inversion or exposure of the IMM, whose surface area greatly exceeds that of the OMM". It is unclear how the behavior of TOMM-20 can lead to this conclusion about the IMM. Some basic controls are missing to evaluate their experimental setup. For example, the loss of mito::GFP and TMRE signals (Fig. 1A and S1A) after wounding may reflect photobleaching rather than a change in membrane permeability or polarization. They assert that laser treatment results in discontinuity in Tomm-20; however, in Fig. 1B, the discontinuity is not obvious compared to control, which also has a region of lower signal. Many of their experiments contain mKate2 localized to mitochondria (Tomm20-Kate2 or mito-Kate2). mKate is known to exhibit a pseudo "photoconversion" from red to green. They should perform controls to formally rule out photoconversion as the source of their green mitochondrial signal. Therefore, some basic aspects of their experimental system need clarification.

A major conclusion is that MAI is a reporter of membrane rupture. If this is the conclusion, there should be independent confirmation. Why would cold stress cause membrane rupture, whereas membrane depolarizing agents would not?

The authors also concluded that laser damaged mitochondria undergo mitophagy. They conclude that "Lysosomes approached and degraded MAI-1-positive mitochondria." [Fig. 4A] It is not clear how this statement can be made. Mitochondria were not imaged before wounding, so approach of lysosomes cannot be evaluated. Degradation of mitochondria seems to be assumed by disappearance of MAI signal; however, that disappearance can be account for by other processes.

Finally, the biological significance of these findings are unclear. They speculate that MAI-1 may act as a surveillance mechanism to detect and degrade damaged mitochondria. For example, it may act to recruit autophagy adaptors or autophagosome components. It should be relatively straightforward to test whether mitochondrial degradation after laser damage is impaired in MAI-1 deficient worms.

Other points:

S1B, C: The area of laser wounding should be indicated in all experiments, but especially in this experiment where the borders are not clear.

(E) The dashed boxes do not appear to be accurately drawn, because the boxed regions do not align with the zoomed images.

2B: Only a few mitochondria are labeled and not robustly. Why is the localization so sparse?

2C: The area of laser illumination needs to be indicated. The authors should explain why the green spots do not completely overlay onto the red tubules.

S3A: Experiment is not described sufficiently. What is bottom panel?

3B and C: Unclear what these plots refer to.

3E and F: Legend does not explain sufficiently

S4A: For needle experiment, only a single spot is evaluated. Need more quantitative analysis. What is the evidence that it is a rupture marker? Why would cold stress cause rupture? Need to show strict correlation with loss of IMS marker or matrix marker.

4B: Arrows should be used to point out the relevant events.

4D: Data indicate that disappearance of MAI signal starts at ~1 hour. It would be good to repeat (C) at a time point shorter than 24 h, like 1 or 2 h.

Reviewer #2 (Comments to the Authors (Required)):

The manuscript by Gao et al. describes MAI-1 selectively labels mitochondria undergoing rupture and proposes a MAI-1/LGG-1-mediated clearance pathway that is independent of PINK1/Parkin. The work is interesting and potentially useful as a tool. The manuscript is novel and of great significance.

1. The authors state that MAI-1 is a human IF1 homologue, but it remains unclear to what extent the conserved domain is preserved.
2. The working model appears to be that MAI-1 recognizes exposed ATP synthase complexes on ruptured mitochondria. If so, is ATP synthase indeed the necessary target for MAI-1 recruitment? The authors are suggested to add discussion.
3. The claim of "millisecond-scale" recruitment should be supported by the actual imaging parameters: frame rate, temporal resolution, and a quantitative analysis of MAI-1 signal changes in the first frames before and after damage.
4. The causal role of MAI-1 in LGG-1 recruitment remains unclear. Suggest to discuss.
5. For each quantitative panel (e.g., Figure 4C), please indicate the sample size n and what n represents in the figure legends.

Several textual and typographical issues should be corrected for clarity and consistency, for example:

"pro-apoptic" → "pro-apoptotic" (Discussion)

"Parkinmediated" → "Parkin-mediated" (Discussion)

"proinflammatory" → "pro-inflammatory" (Discussion)

"damageappropriate" → "damage-appropriate" (Discussion)

"agr-1" → "argn-1" (Figure 2B)

Use a consistent notation for the reporter: "MAI1::GFP" and "MAI-1::GFP".

"MAI-1 translocation was independently of tag orientation..." → "independent of tag orientation..." (Results).

In Figure S4C, "fco-1(tm1133)" should be corrected to "fzo-1(tm1133)".

Reviewer #3 (Comments to the Authors (Required)):

In this paper, the authors identified MAI-1 as a genetically encoded reporter of mitochondrial membrane rupture. The authors found that MAI-1 is targeted to mitochondria that undergo catastrophic and irreversible rupture rather than reversible bioenergetic fluctuations. Finally, they demonstrated that MAI-1-targeted damaged mitochondria are removed by LGG-1 and ATG-3-mediated degradation in a PINK-1 and PDR-1/PARKIN-independent manner. Although the data are of high quality and

contain interesting results, several points are not sufficiently supported, and additional data or further explanation are necessary. In particular, physiological roles of MAI-1 in the elimination of damaged mitochondria should be studied in detail.

Major comments

- 1) The authors should examine the phenotypes of *mai-1* knockout mutants in detail. Does MAI-1 loss aggravate further damaged mitochondrial structures? Does its loss affect LGG-1 localization or damaged mitochondrial clearance? In addition, a basic analysis of *mai-1* knockout mutants should be conducted (brood size, lifespan, motility, etc.).
- 2) The expression pattern and subcellular localization of endogenous MAI-1 proteins should be examined.
- 3) The authors should use SIM to clarify whether MAI-1 localizes to the inner mitochondrial membrane or the matrix of damaged mitochondria.
- 4) When GFP alone is expressed under the same conditions, is it also targeted to the mitochondria?
- 5) Fi. S3C. Laser wounding caused excessive fading of mKate2, making it difficult to detect signals within or near the mitochondria.
- 6) Is there any IF1 homolog lacking MTS like MAI-1 in other organisms?
- 7) The reason the authors focused on MAI-1 is unclear and needs to be explained. If there are other mitochondrial-associated proteins, such as MAI-1, that lack MTS, please provide a list.
- 8) It would be better to statistically analyze the colocalization of MAI-1 and mitochondria in each Figure.

Minor comments

1) Fig.2D indicates "Tomm-20::Yonghong" but its legend describes "TOMM-20::mKate2". In several figures, it is written as "Tomm-20::mKate2," but it should be corrected to "TOMM-20::mKate2."

2) Fig. 4D. LGG-1::GFP should be corrected to GFP::LGG-1.

3) Methods

Appropriate papers should be cited in the following sentences.

(p9). "RNAi was performed as previously described (reference). "

(p10) "Colocalization analysis

The colocalization analysis was as previously described (reference, The Golgi apparatus facilitates membrane repair). "

Because there are no data for colocalization analysis, the authors should reconsider whether this chapter is necessary.

4) The 5 min and 24 h images in Figure 4C and Figure S6B (although S6 has poor image quality) appear to be identical; therefore, Figure S6B may not be necessary. In addition, the legend for Figure S6B mentions 1 h and 24 h, but there is no 1 h image.

5) In the main text, the *pdr-1* allele is listed as *tm598*, but in the Figures it is *gk448*. In addition, the allele name for *atg-3* in Figure 4C is indicated as *tbp412*, but it is described as *bp412* in the legend. Some strains used in this study (e.g., *atg-3* mutant) are not listed in supplemental Table 1.

6) In several cases, fluorescently-tagged proteins are indicated as "MAI-1-GFP". These should be corrected to MAI-1::GFP.

7) The construction methods of plasmids and strains should be described in more detail. For P_{col-19}::MAI-1::FP, please specify which part of the *col-19* promoter was used. In addition, the origin of the fluorescent protein fragments and whether they were optimized for *C. elegans* should be mentioned. Additionally, some plasmids in Sup Table 2 lack details on how they were constructed.

Reviewer comments (our point-by-point responses are included in blue text):

Reviewer #1 (Comments to the Authors (Required)):

The study by Gao et al identifies IF1 as a sensor of mitochondrial membrane rupture. They show that the cytosolic IF1 in *C. elegans* localizes to mitochondria after laser damage and some other types of mitochondrial damage. This property can be engineered with human IF1 by removing the mitochondrial targeting sequence. They suggest that this provides a new tool to study mitochondrial physiology.

Overall, the conclusion of the study is interesting, because it indeed would be useful to have a genetically encoded reporter of mitochondrial membrane integrity. However, the conclusions made by the authors need more supporting evidence to be fully convincing.

The study relies entirely on a single technique--the imaging of fluorescent reporters after a perturbation. This limits their ability to interpret the physical basis of their observation, and some of their statements appear overinterpreted. Second paragraph of the Results says that "After wounding, TOMM-20::mKate2 became discontinuous surrounding the ATP-3::GFP (Figure 1B), consistent with inversion or exposure of the IMM, whose surface area greatly exceeds that of the OMM". It is unclear how the behavior of TOMM-20 can lead to this conclusion about the IMM. Some basic controls are missing to evaluate their experimental setup. For example, the loss of mito::GFP and TMRE signals (Fig. 1A and S1A) after wounding may reflect photobleaching rather than a change in membrane permeability or polarization. They assert that laser treatment results in discontinuity in Tomm-20; however, in Fig. 1B, the discontinuity is not obvious compared to control, which also has a region of lower signal. Many of their experiments contain mKate2 localized to mitochondria (Tomm20-Kate2 or mito-Kate2). mKate is known to exhibit a pseudo "photoconversion" from red to green. They should perform controls to formally rule out photoconversion as the source of their green mitochondrial signal. Therefore, some basic aspects of their experimental system need clarification.

A major conclusion is that MAI is a reporter of membrane rupture. If this is the conclusion, there should be independent confirmation. Why would cold stress cause membrane rupture, whereas membrane depolarizing agents would not?

The authors also concluded that laser damaged mitochondria undergo mitophagy. They conclude that "Lysosomes approached and degraded MAI-1-positive mitochondria." [Fig. 4A] It is not clear how this statement can be made. Mitochondria were not imaged before wounding, so approach of lysosomes cannot be evaluated. Degradation of mitochondria seems to be assumed by disappearance of MAI signal; however, that disappearance can be accounted for by other processes.

Finally, the biological significance of these findings are unclear. They speculate that MAI-1 may act as a surveillance mechanism to detect and degrade damaged mitochondria. For example, it may act to recruit autophagy adaptors or autophagosome components. It should be relatively straightforward to test whether mitochondrial degradation after laser damage is impaired in MAI-1 deficient worms.

Response:

We thank the reviewer for their thoughtful and constructive evaluation of our work. We appreciate the recognition of the value of MAI-1/IF1 as a genetically encoded reporter of mitochondrial membrane integrity. We agree that several conclusions in the original submission required clearer phrasing, additional controls, and more direct supporting evidence. In the revised manuscript, we have substantially refined our interpretations, performed new experiments, and toned down a few overstatements to better align conclusions with the data.

1. Interpretation of TOMM-20 and ATP-3 discontinuity after laser wounding

We agree that our original statement suggesting “inversion or exposure of the IMM, whose surface area greatly exceeds that of the OMM” based on TOMM-20 behavior, was overinterpreted. In the revised manuscript, we have replaced this sentence with a more conservative description: “*After wounding, TOMM-20::mKate2 became fragmented and discontinuous around the ATP-3::GFP at the damage site (Figure 1D), indicating disruption of mitochondrial structural integrity*”. We describe the imaging results strictly as loss of continuity of both outer and inner mitochondrial membrane markers, consistent with structural disruption of mitochondria following laser injury. We have also revised Figure 1 to better illustrate these discontinuities and clarified them in the text.

2. Photobleaching and loss of mito::GFP and TMRE signals

We appreciate the concern that loss of fluorescence signals could reflect photobleaching. To address this, we performed additional control experiments, now included in the revised manuscript. Time-lapse imaging under laser exposure in mito::GFP and TMRE signals were lost specifically at the wound site but not in the surrounding regions (new Figure 1A). At the same time, ATP-3::GFP and TOMM-20::GFP remains and TMRE signals are lost at the wound site (new Figure 1B, 1C, and new Figure S1B, S1C), consistent with loss of membrane potential accompanying structural disruption rather than photobleaching alone. We have clarified this point in the Results and Figure legends.

3. Potential photoconversion of mKate2

We agree that mKate2-family fluorophores can exhibit red-to-green photoconversion under certain imaging conditions. To rule out photoconversion as the source of the green mitochondrial signal, we performed wounding on TOMM-20::mKate2 alone. Under these conditions, no GFP-like signal appeared at damaged mitochondria (new Figure S2F), suggesting that mKate2 does not photoconvert into a green fluorophore that selectively localizes to the damaged site.

In addition, we performed co-labeling experiments using MAI-1::mKate2 together with ATP-3::GFP. After mitochondrial damage, red MAI-1::mKate2 robustly accumulated on ATP-3::GFP-labeled mitochondria (new Figure 2G). This independent green mitochondrial marker confirms that MAI-1::mKate2 undergoes genuine damage-induced mitochondrial recruitment rather than artifactual signal conversion.

Together, these controls demonstrate that the observed MAI-1 translocation is not attributable to fluorophore photoconversion. These data are now included in the revised manuscript.

4. Evidence that MAI-1 reports mitochondrial membrane rupture

We agree that independent validation is important. We now explicitly describe MAI-1 as a putative sensor of mitochondrial membrane damage, supported by multiple injury paradigms, including laser-induced damage, needle wounding, ultrasonic-induced wounding, cold stress, miniSOG-induced specific mitochondria damage, and also genetically mitochondrial membrane damage mutants (new Figure 4, Figure S3).

Importantly, several negative controls demonstrate the specificity of MAI-1 recruitment. Cytosolic GFP does not accumulate at damaged mitochondria (new Figure S2C), and truncated MAI-1 variants fail to localize to injury sites (new Figure 3E–3G), indicating that MAI-1 accumulation is not due to passive diffusion or nonspecific aggregation.

Together, these complementary approaches support the conclusion that MAI-1 selectively accumulates on mitochondria with compromised membrane integrity and functions as a putative reporter of mitochondrial membrane rupture.

5. Cold stress versus mitochondrial depolarizing agents

We appreciate the reviewer's question regarding why cold stress induces MAI-1 recruitment, whereas membrane depolarizing agents do not. We have added a brief mechanistic explanation and supporting reference in the revised discussion. Extensive literature demonstrates that freezing or subzero stress induces mechanical membrane damage, osmotic stress, and ROS production that can physically compromise mitochondrial membranes. Multiple studies have documented mitochondrial swelling,

membrane rupture, or permeability transition under freezing stress in diverse systems (DOI: 10.1002/bit.21981, 10.1098/rspb.2020.1273, 10.3389/fphys.2024.1358190).

In contrast, FCCP/CCCP, oligomycin, and rotenone primarily dissipate membrane potential or inhibit respiration without necessarily causing immediate physical membrane rupture. While these treatments efficiently induce mitophagy, they do not invariably produce catastrophic membrane structural rupture. Consistent with this distinction, our data support the conclusion that MAI-1 specifically senses the membrane rupture rather than reversible mitochondrial depolarizations.

6. Lysosomal engagement of MAI-1-positive mitochondria.

We agree that our original wording implied lysosomal degradation without sufficient direct evidence. To address this concern, we performed new live-imaging experiments to more directly examine lysosomal involvement following mitochondrial damage.

First, SCAV-3::GFP-labeled lysosomal membranes were imaged together with MAI-1::mKate2 in time-lapse imaging (new Figure 8D and new Figure S5C). These movies show lysosomal membranes dynamically approaching and enveloping MAI-1-positive mitochondria following laser damage. In addition, the MAI-1::mKate2 signal became enclosed with SCAV-3::GFP-positive structure and gradually decreased in size and intensity, consistent with the putative lysosomal degradation process.

Second, we performed triple labeling of TOMM-20::mYonghong (outer mitochondrial membrane), NUC-1::mNeonGreen (lysosomal lumen), and MAI-1::BFP. Line-scan analysis demonstrates spatial convergence of mitochondria, lysosomes, and MAI-1 signals after injury (new Figure 8C). Notably, MAI-1::BFP labeled damaged mitochondria, but not surrounding TOMM-20::mYonghong-labeled intact mitochondria, colocalized with lysosome lumen marker NUC-1::mNeonGreen, suggesting selective recruitment of damaged mitochondria by lysosome.

Based on the reviewer's guidance and these new data, together with our original findings that MAI-1::GFP is associated with NUC-1::mCherry at the wound site (new Figure 8A and 8B), we have now revised the manuscript to state more cautiously that lysosomes *engage* and *engulf* MAI-1-positive mitochondria after damage.

7. Biological significance and function of MAI-1

We agree that functional analysis of MAI-1 in mitochondrial quality control represents an important future direction. In the revised Discussion, we frame MAI-1 as a potential component of a mitochondrial damage surveillance pathway and outline future experiments that will be required to determine whether mitochondrial clearance after injury is impaired in MAI-1-deficient animals.

As advised by the editor, we have limited exploration of MAI-1's endogenous physiological roles in mitochondrial homeostasis and stress responses in this manuscript. Accordingly, we present MAI-1 primarily as a genetically encoded reporter of mitochondrial membrane rupture, while explicitly acknowledging that elucidating its full physiological functions will be an important subject of future investigation.

Other points:

S1B, C: The area of laser wounding should be indicated in all experiments, but especially in this experiment where the borders are not clear.

We thank the reviewer for this suggestion. In the revised figures, we have added a white dashed circle to indicate the laser-wounded area in all relevant panels, including S1B and S1C, to clearly delineate the illuminated region experiments.

(E) The dashed boxes do not appear to be accurately drawn, because the boxed regions do not align with the zoomed images.

We apologize for this oversight. The dashed boxes have been redrawn to align precisely with the corresponding zoomed regions in the revised figures.

2B: Only a few mitochondria are labeled and not robustly. Why is the localization so sparse?

We appreciate this insightful question. The *aass-1(yq211)* and *argn-1(yq187)* mutants disrupt mitochondrial homeostasis through accumulation of specific metabolites (saccharopine or arginine, respectively) within mitochondria, as reported previously (DOI: 10.1083/jcb.201807204 and 10.1016/j.jgg.2020.02.007). These metabolic perturbations do not directly rupture membranes but instead impair mitochondrial dynamics and function, leading to heterogeneous mitochondrial damage. Consistent with published work and their transmission electron microscopy, only a subset of mitochondria displays pronounced swelling or structural disruption in our experiment.

We therefore interpret the sparse MAI-1::GFP labeling as reflecting selective accumulation on the subset of metabolically damaged mitochondria.

2C: The area of laser illumination needs to be indicated. The authors should explain why the green spots do not completely overlay onto the red tubules.

We thank the reviewer for this helpful suggestion. The miniSOG-induced damage was exposed to blue light on the plate, and the entire animal can be illuminated. Thus, we did not indicate the illumination in the figures, but we describe this in detail in the Methods.

We also expanded the Methods and Results to clarify that mitochondria-targeted miniSOG generates singlet oxygen locally at the outer mitochondrial membrane upon blue-light illumination (DOI: 10.1038/srep21271 and 10.1073/pnas.1204096109). This localized production of singlet oxygen induces sustained oxidative cause server damage to mitochondria, including fragmentation, quenches or reduces the TOMM-20::mKate2 fluorescence. We have added line-scan analyses showing that the MAI-1::GFP signal remains spatially coincident with mitochondria despite partial reduction of the TOMM-20::mKate2 signal (new Figure 4C).

S3A: Experiment is not described sufficiently. What is bottom panel?

We apologize for the confusion caused by the unclear description. The original bottom panel, shown inverted white and black, is the same image as the top panel. We now removed the bottom panel in the new revised Figures.

3B and C: Unclear what these plots refer to.

We apologize for the lack of clarity. We have rewritten the Results text and figure legends to clearly describe the plotted parameters, experimental conditions, and quantification methods (new Figure 5B and 5C).

3E and F: Legend does not explain sufficiently

We have revised the figures by providing information of quantification area from panel D and describe detail in the legends (new Figure 5E and 5F).

S4A: For needle experiment, only a single spot is evaluated. Need more quantitative analysis. What is the evidence that it is a rupture marker? Why would cold stress cause rupture? Need to show strict correlation with loss of IMS marker or matrix marker.

We appreciate the reviewer's important points regarding quantification and validation of membrane rupture.

Regarding the needle wounding experiments, prior studies of mechanical injury in *C. elegans* epidermis have shown that needle puncture generates a highly localized wound core in which organelles and cytoskeletal structures are physically displaced or destroyed, leaving only a narrow peripheral zone containing residual mitochondria (DOI: 10.1038/s41467-020-14885-x and 10.1016/j.devcel.2014.08.002). Consistent with this, only a limited number of mitochondria are detectable near the wound margin and thus available for MAI-1::GFP labeling. We have clarified this rationale explicitly in the revised manuscript and added the appropriate references.

To strengthen the evidence that MAI-1 reports mitochondrial membrane rupture rather than nonspecific stress, we now emphasize that MAI-1 recruitment is observed only at mitochondria immediately adjacent to the wound site and not in surrounding intact regions, consistent with a spatially restricted physical insult rather than a global response. We have also added quantitative analysis of MAI-1-positive mitochondria in the wound-adjacent region versus distal regions (new Figure S3A).

Regarding cold stress, which has been shown to induce rapid mechanical stress and oxidative damage to mitochondrial membranes, particularly in ectothermic organisms, leading to structural compromise rather than reversible depolarization. As noted in our response above, we now describe cold stress as a condition that can induce mitochondrial membrane damage, rather than asserting rupture as a universal outcome, and we cite relevant literature supporting this interpretation. Please see also our previous response in the main part.

To directly address the reviewer's comments, we performed new experiments using animals co-expressing the OMM marker TOMM-20::mKate2 and the mitochondrial matrix marker mito::GFP under cold stress. Confocal imaging revealed selective loss of the matrix marker mito::GFP, while the TOMM-20::mKate2 signal was retained. These data demonstrate that cold shock causes matrix loss while still retaining membrane architecture. The new data are included and described in the revised Results and Figures S3B.

Together, these additional analyses and clarifications strengthen the conclusion that MAI-1 preferentially associates with mitochondria whose membrane integrity is compromised and support its use as a putative reporter of mitochondrial membrane rupture.

4B: Arrows should be used to point out the relevant events.

We thank the reviewer for this suggestion. We have added the white arrows to point out the relevant areas in time-lapse images (new Figure 8B).

4D: Data indicate that disappearance of MAI signal starts at ~1 hour. It would be good to repeat (C) at a time point shorter than 24 h, like 1 or 2 h.

We agree and have now included an additional time point at 2 hours post-wounding, showing early-stage reduction of MAI-1 signal (new Figure 7B and 7C). These data better illustrate the temporal progression of MAI-1-positive mitochondria following damage.

Reviewer #2 (Comments to the Authors (Required)):

The manuscript by Gao et al. describes MAI-1 selectively labels mitochondria undergoing rupture and proposes a MAI-1/LGG-1-mediated clearance pathway that is independent of PINK1/Parkin. The work is interesting and potentially useful as a tool. The manuscript is novel and of great significance.

1. The authors state that MAI-1 is a human IF1 homologue, but it remains unclear to what extent the conserved domain is preserved.

We thank the reviewer for this insightful suggestion. In the revised manuscript, we have added multiple sequence alignments among *C. elegans* MAI-1, MAI-2, and human IF1, together with a percent identity matrix comparing MAI-1 and MAI-2 to human IF1 (Figure 2A, Figure 3A, and Figure 3B). These analyses show that MAI-1 preserves the core IF1 inhibitory domain, including residues implicated in ATP synthase interaction, supporting its designation as an IF1 homolog. We have included these data and described them in the Results.

2. The working model appears to be that MAI-1 recognizes exposed ATP synthase complexes on ruptured mitochondria. If so, is ATP synthase indeed the necessary target for MAI-1 recruitment? The authors are suggested to add discussion.

We thank the reviewer for these excellent suggestions. Human IF1 binds the interface between the α and β subunits of ATP synthase, suggesting ATP synthase as a plausible interaction partner exposed upon mitochondrial membrane rupture. Consistent with this, we observed strong colocalization of MAI-1 with ATP-3 at mitochondrial injury sites (new Figure 6A).

To directly test whether ATP synthase binding is necessary for MAI-1 recruitment, we performed RNAi knockdown of essential components of ATP synthase and quantified MAI-1::GFP accumulation at laser-damaged mitochondria. In the revised manuscript, we now show that both *atp-1* and *atp-3* RNAi abolishes MAI-1::GFP accumulation at wound sites at both 1 min and 5 min post-damage compared with controls (new Figure 6C and 6D). ATP-1 is the *C. elegans* ortholog of human ATP5F1A (ATP synthase F1 subunit alpha), while ATP-3 is the ortholog of human ATP5PO. These results demonstrate that the presence of an intact ATP synthase complex is required for efficient MAI-1 recruitment, supporting ATP synthase complex as a likely binding target mediating the rapid translocation of MAI-1 to ruptured mitochondria. We have now incorporated these data into the revised manuscript.

We thank the reviewer for this insightful suggestion, which enabled us to mechanistically clarify how MAI-1 is recruited to damaged mitochondria.

3. The claim of "millisecond-scale" recruitment should be supported by the actual imaging parameters: frame rate, temporal resolution, and a quantitative analysis of MAI-1 signal changes in the first frames before and after damage.

We thank the reviewer for this important suggestion. In the revised manuscript, we now provide full imaging parameters (frame rate, exposure time, laser setting, and EM gain) in the Methods. We also added new high-temporal-resolution time-lapse imaging acquired at 200 ms intervals (new Figure 2D). Quantification of MAI-1::GFP intensity in the first frames before and after damage demonstrates that MAI-1::GFP accumulation at damaged mitochondria is detectable within 200 ms of laser injury. We have therefore retained the term "sub-second" recruitment and provide quantitative support for this statement.

4. The causal role of MAI-1 in LGG-1 recruitment remains unclear. Suggest to discuss.

We agree that the mechanistic relationship between MAI-1 and LGG-1 requires careful interpretation. How LGG-1 is recruited to ruptured mitochondria remains an open question.

LGG-1 does not bind mitochondria directly but marks autophagosomal membranes that engage damaged organelles. Our data show that clearance of MAI-1 marked mitochondria occurs independently of PINK-1/PDR-1, indicating a rupture-activated autophagy pathway distinct from canonical mitophagy. Physical membrane disruption may expose normally sequestered inner membrane components or lipids, such as cardiolipin, or promote local autophagosome nucleation near the damaged mitochondria. Although MAI-1 robustly marks ruptured mitochondria before LGG-1 recruitment, whether MAI-1 directly participates in autophagy initiation or functions solely as a damage marker will require future genetic and biochemical analyses. We have revised the Discussion accordingly to reflect these points.

5. For each quantitative panel (e.g., Figure 4C), please indicate the sample size n and what n represents in the figure legends.

We have now added sample sizes and definitions of n to all quantitative figure legends (e.g., new Figure 7C), specifying the number of animals or mitochondria analyzed per condition.

Several textual and typographical issues should be corrected for clarity and consistency, for example:

"pro-apoptic" → "pro-apoptotic" (Discussion)

"Parkinmediated" → "Parkin-mediated" (Discussion)

"proinflammatory" → "pro-inflammatory" (Discussion)

"damageappropriate" → "damage-appropriate" (Discussion)

"agrn-1" → "argn-1" (Figure 2B)

All suggested corrections have been made. We thank the reviewer for noting these errors.

Use a consistent notation for the reporter: "MAI1::GFP" and "MAI-1::GFP".

Corrected.

"MAI-1 translocation was independently of tag orientation..." → "independent of tag orientation..." (Results).

Corrected.

In Figure S4C, "fco-1(tm1133)" should be corrected to "fzo-1(tm1133)".

Corrected.

Reviewer #3 (Comments to the Authors (Required)):

In this paper, the authors identified MAI-1 as a genetically encoded reporter of mitochondrial membrane rupture. The authors found that MAI-1 is targeted to mitochondria that undergo catastrophic and irreversible rupture rather than reversible bioenergetic fluctuations. Finally, they demonstrated that MAI-1-targeted damaged mitochondria are removed by LGG-1 and ATG-3-mediated degradation in a PINK-1 and PDR-1/PARKIN-independent manner. Although the data are of high quality and contain interesting results, several points are not sufficiently supported, and additional data or further explanation are necessary. In particular, physiological roles of MAI-1 in the elimination of damaged mitochondria should be studied in detail.

We thank the reviewer for the careful evaluation of our manuscript and for the positive assessment of data quality and novelty. We appreciate the insightful comments and suggestions. In response, we have performed additional experiments, clarified interpretations, and moderated conclusions where appropriate. Below, we address each point in detail.

Major comments

1) The authors should examine the phenotypes of *mai-1* knockout mutants in detail. Does MAI-1 loss aggravate further damaged mitochondrial structures? Does its loss affect LGG-1 localization or damaged mitochondrial clearance? In addition, a basic analysis of *mai-1* knockout mutants should be conducted (brood size, lifespan, motility, etc.).

We fully agree that understanding the physiological role of MAI-1 is important. As *C. elegans* contains two IF1 homologies, MAI-1 (lacking MTS) and MAI-2 (with MTS), this adds complexity to interpreting their distinct function. As advised by the editor, this manuscript will be considered as a Tool format for the journal; therefore, a comprehensive analysis of the physiological role of MAI-1 in mitochondrial

homeostasis and mitochondrial clearance will be an important direction for future studies. We will study their function in the future.

2) The expression pattern and subcellular localization of endogenous MAI-1 proteins should be examined.

We thank the reviewer for this suggestion. We generated a transcriptional and translational reporter using a 760 bp promoter region upstream of the *mai-1* start codon fused to MAI-1::GFP. Single-copy insertion animals show MAI-1 expression in multiple tissues, including neurons, pharynx, hypodermis, body wall muscle, and coelomocytes (new Figure S2B).

Under basal conditions, MAI-1::GFP displays diffuse cytosolic localization, consistent with our *Pcol-19*::MAI-1::GFP reporter. These data are now included in a new figure and described in the Results.

3) The authors should use SIM to clarify whether MAI-1 localizes to the inner mitochondrial membrane or the matrix of damaged mitochondria.

We appreciate this suggestion. We have now performed structured illumination microscopy (SIM) imaging of MAI-1::BFP together with ATP-3::GFP (inner mitochondrial membrane marker) and TOMM-20::mYonghong (outer mitochondrial membrane marker). Single optical sections show stronger spatial overlap of MAI-1::BFP with ATP-3::GFP than with TOMM-20::mYonghong (new Figure 6B). These data suggest that MAI-1 accumulates on or near the inner mitochondrial membrane following damage. We now describe this as evidence consistent with inner membrane association, while noting that higher-resolution methods will be needed to determine precise topology.

4) When GFP alone is expressed under the same conditions, is it also targeted to the mitochondria?

We agree that this is an important control. We generated animals expressing *Pcol-19*::GFP alone and performed time-lapse imaging following laser injury. Diffuse GFP remains cytosolic and does not accumulate on damaged mitochondria (new Figure S2C). These results confirm that mitochondrial recruitment is specific to MAI-1 and not a property of free GFP.

5) Fi. S3C. Laser wounding caused excessive fading of mKate2, making it difficult to detect signals within or near the mitochondria.

We apologize for the unclear presentation. We have replaced these images with improved acquisitions showing reduced mKate2 fading. In addition, we performed line-scan analyses of truncated MAI-1::mKate2 together with TOMM-20::GFP (new Figure 3F and 3G), demonstrating that truncated MAI-1 constructs do not accumulate at damaged mitochondria.

6) Is there any IF1 homolog lacking MTS like MAI-1 in other organisms?

We thank the reviewer for this interesting question. We have added a brief note that most characterized IF1 proteins contain canonical or cryptic mitochondrial targeting sequences (new Figure S2G). However, most nematode species, as well as *Drosophila* and zebrafish, possess MAI-1-like IF1 proteins that naturally lack an MTS, although such variants are not reported in humans and mice (new Figure S2G). These suggest that the MTS-deleted version of IF1 can be engineered in other systems to visualize mitochondrial membrane rupture, providing a potential strategy to study mitochondrial membrane integrity across species.

7) The reason the authors focused on MAI-1 is unclear and needs to be explained. If there are other mitochondrial-associated proteins, such as MAI-1, that lack MTS, please provide a list.

We appreciate this request for clarification. We have expanded to explain that *mai-1* was initially identified in our previous work as a wound-responsive gene in an MS2-based live RNA imaging screen (DOI: 10.7554/eLife.82178). This motivated us to examine MAI-1 protein dynamics, which unexpectedly revealed rapid mitochondrial recruitment upon damage.

While the existence of other MAI-1-like proteins lacking classical mitochondrial targeting sequences is not fully known, several mitochondrial-associated proteins that lack classical N-terminal targeting sequences can be recruited under specific conditions via alternative mechanisms, including protein-protein interactions (DRP-1, etc.), damage-induced recruitment (PARKIN, P62, etc.), or covalent-heme-anchoring (Cytochrome C, etc.).

A List of proteins that are associated with mitochondria without MTS

Types	Protein
Protein-protein interactions	DRP1, HK2

Stress-induced recruitment	PARKIN, p62, OPTN, NDP52, NBR1,
Covalent-heme-anchoring	Cytochrome c

We emphasize that MAI-1 is distinct from these proteins because it is a compact IF1-family protein that is soluble under basal conditions and rapidly and specifically translocates to mitochondria upon mitochondrial membrane rupture, functioning as a genetically encoded reporter of catastrophic mitochondrial damage.

8) It would be better to statistically analyze the colocalization of MAI-1 and mitochondria in each Figure.

We thank the reviewer for this suggestion. We have now quantified colocalization using Pearson's correlation coefficients for MAI-1, cytosolic MAI-2, and cytosolic human IF1 reporters relative to mitochondrial markers. These analyses confirm a significant correlation, specifically after damage. These data are included in a new figure and described in the Results (new Figure 2F, 2G, 3C, and 3D).

Minor comments

1) Fig.2D indicates "Tomm-20::Yonghong" but its legend describes "TOMM-20::mKate2". In several figures, it is written as "Tomm-20::mKate2," but it should be corrected to "TOMM-20::mKate2."

Corrected.

2) Fig. 4D. LGG-1::GFP should be corrected to GFP::LGG-1.

Corrected.

3) Methods

Appropriate papers should be cited in the following sentences.

(p9). "RNAi was performed as previously described (reference). "

Corrected.

(p10) "Colocalization analysis

The colocalization analysis was as previously described (reference, The Golgi apparatus facilitates membrane repair). "

Because there are no data for colocalization analysis, the authors should reconsider whether this chapter is necessary.

Thanks for the suggestions. We have now added colocalization analysis to most of our colocalization images show MAI-1 localized to mitochondria after damage.

4) The 5 min and 24 h images in Figure 4C and Figure S6B (although S6 has poor image quality) appear to be identical; therefore, Figure S6B may not be necessary. In addition, the legend for Figure S6B mentions 1 h and 24 h, but there is no 1 h image.

We appreciate the reviewer's careful observations. We have now removed this figure and corrected the figure organization accordingly. In addition, we have reorganized the figures throughout the manuscript in response to reviewer's constructive comments.

5) In the main text, the *pdr-1* allele is listed as tm598, but in the Figures it is gk448. In addition, the allele name for *atg-3* in Figure 4C is indicated as *tbp412*, but it is described as *bp412* in the legend. Some strains used in this study (e.g., *atg-3* mutant) are not listed in supplemental Table 1.

Corrected.

6) In several cases, fluorescently-tagged proteins are indicated as "MAI-1-GFP". These should be corrected to MAI-1::GFP.

Corrected.

7) The construction methods of plasmids and strains should be described in more detail. For *Pcol-19::MAI-1::FP*, please specify which part of the *col-19* promoter was used. In addition, the origin of the fluorescent protein fragments and whether they were optimized for *C. elegans* should be mentioned. Additionally, some plasmids in Sup Table 2 lack details on how they were constructed.

We appreciate the reviewer's suggestion and have revised the Methods and Supplementary Tables 2 and 3 to provide additional detail on plasmid and strain construction. Specifically, we now specify the fragment of the *col-19* promoter used to generate the *Pcol-19::MAI-1::FP* construct, describe the origin of all fluorescent protein sequences, and indicate whether they were codon-optimized for *C. elegans*. In addition, we have clarified the construction strategies for plasmids that previously lacked sufficient methodological detail in Supplementary Table 2.

We sincerely thank the reviewers for their careful evaluation and for bringing these important issues to our attention. We are particularly grateful for the insightful feedback,

which has significantly improved the quality of this study. We believe that the revisions made in response to their comments have substantially strengthened the rigor and clarity of the manuscript.

March 10, 2026

Re: JCB manuscript #202511088R

Suhong Xu
Zhejiang University

Dear Prof. Xu,

Thank you for submitting your revised manuscript entitled "A cytosolic IF1 enables real-time visualization of mitochondrial membrane rupture". The manuscript has been seen by the original reviewers whose full comments are appended below. While the reviewers continue to be overall positive about the work in terms of its suitability for JCB, some important issues remain.

You will see that reviewer #1 has serious remaining concerns about there being insufficient evidence for use of MAI-1 as a reporter of membrane rupture. To ensure that MAI-1 functions as a reporter for ruptured mitochondria, and that its use does not lead to potential confusion in the field, this issue must be completely resolved with new data as requested. However, as stipulated in our initial editorial letter, determining the physiological function of MAI-1 as requested by reviewer #3 is beyond the requirements for a JCB Tool.

Our general policy is that papers are considered through only one revision cycle; however, we are open to one additional short round of revision to conclusively address this issue. Please note however that no further revisions will be permitted.

Please submit the final revision within one month (or let us know if additional time is necessary), along with a cover letter that includes a point by point response to the remaining reviewer comments.

Thank you for this interesting contribution to Journal of Cell Biology. You can contact me or the scientific editor listed below at the journal office with any questions at cellbio@rockefeller.edu.

Sincerely,

Hong Zhang, PhD
Monitoring Editor

Andrea L. Marat, PhD
Deputy Editor

Journal of Cell Biology

Reviewer #1 (Comments to the Authors (Required)):

The revised manuscript continues to have notable concerns about the setup of the study. In Figure 1, TOMM20-mKate2 stays somewhat intact, which the authors interpret as maintenance of mitochondrial structure despite loss of matrix contents. However, panel D shows rapid loss of TOMM20-mYonghong, which is interpreted as disruption of structural integrity. It is hard to reconcile the different behaviors of the similar fluorescent proteins (TOMM20-mKate2 and TOMM20-mYonghong), and the drastically different interpretations. Figure 1 and S1 do not rigorously address the issue of photobleaching vs rupture of the mitochondrial membranes, which is a foundational distinction for this study. Fig. S1A show differential loss of matrix vs OM or IM GFPs, but it cannot be ruled out that the different GFPs, due to their fusion to partners, have different sensitivities to laser light. In the matrix experiments, can it be shown that the surrounding region (outside the mitochondria) gains fluorescence upon wounding, as would be expected for rupture of mitochondria and leakage (vs photobleaching) of GFP? With the current results, which rely solely on imaging of fluorescent proteins, it is uncomfortable to conclude that their laser treatment results in membrane rupture.

It seems that MAI-1 translocates to mitochondria upon various stresses, but it seem premature to conclude that membrane rupture is the relevant parameter. The authors assume that the various stresses used (cold, genetic mutations) result in rupture.

The authors assume that MAI-1 functions as a reporter of ruptured mitochondria, and maybe even as a facilitator of subsequent mitochondrial degradation. The known function of IF1 is to inhibit the hydrolytic activity of ATP synthase. A simpler model is that, in nematodes, MAI-1 functions to shut down ATP synthase upon membrane rupture. The translocation of cytosolic MAI-1 to mitochondria is simply a consequence of its ability to access its binding partner, which is embedded in the IM and is normally inaccessible. Ruptured mitochondria cannot effectively provide ATP generation for the cell, and MAI-1 may function to inhibit

ATP synthase under such pathological conditions. The subsequent mitochondrial degradation is probably taken care of by other machinery and may have nothing to do with MAI-1. This can be tested by examining mitochondrial degradation in an MAI-1 knockdown or mutant, but the authors argue that this is outside the scope of the study.

The logic for the following conclusion is unclear: "Notably, although endogenous MTS-lacking IF1 proteins have not been identified in mammals, engineered cytosolic IF1 variants from diverse species display conserved rupture-induced mitochondrial recruitment, suggesting that membrane structural damage surveillance is an evolutionarily conserved property of IF1-family proteins. This raises the possibility that analogous mechanisms operate in mammalian tissues, where mitochondrial membrane rupture contributes to neurodegeneration, ischemia-reperfusion injury, and inflammatory pathologies." In fact, the authors' data clearly argue against such a mechanism in mammals or most organisms. They show that such a mechanism is plausible in nematodes (with the caveat in the paragraph above), but not used in mammals.

Since the authors focus on the potential technological utility of the MAI-1 rather than its biological function, they should elaborate on the utility of designing a reporter for catastrophic mitochondrial rupture. The study suggests that many types of rather severe mitochondrial stress, including loss of membrane potential or various genetic mutants, do not recruit MAI-1. MAI-1 recruitment may be limited to very harsh stresses that likely result in broad cellular damage. It would be helpful to point out what specific types of studies would be facilitated by such a reporter.

Reviewer #2 (Comments to the Authors (Required)):

The authors have answered my questions and I suggest acceptance.

Reviewer #3 (Comments to the Authors (Required)):

In the revised manuscript, the authors addressed some of my comments and those of other reviewers. However, the authors have not performed fundamental analyses using the *mai-1* knockout mutant to demonstrate the physiological function of MAI-1 in the removal of damaged mitochondria. As previously pointed out in Major Comment 1, a detailed analysis of MAI-1 function in this process is considered essential for this study.

Reviewer comments (our point-by-point responses are included in blue text):

Reviewer #1 (Comments to the Authors (Required)):

The revised manuscript continues to have notable concerns about the setup of the study. In Figure 1, TOMM20-mKate2 stays somewhat intact, which the authors interpret as maintenance of mitochondrial structure despite loss of matrix contents. However, panel D shows rapid loss of TOMM20-mYonghong, which is interpreted as disruption of structural integrity. It is hard to reconcile the different behaviors of the similar fluorescent proteins (TOMM20-mKate2 and TOMM20-mYonghong), and the drastically different interpretations.

Response:

We thank the reviewer for this important and insightful comment, which prompted us to carefully re-evaluate our data and improve both the rigor and clarity of the manuscript. We sincerely apologize for the confusion caused by our previous presentation and agree that the apparent discrepancy between TOMM-20::mKate2 (Figure 1B) and TOMM-20::mYonghong (original Figure 1D) required clarification, which should have been done during the previous revision.

Based on our systematic re-analysis, we conclude that the observed differences do not reflect a biological inconsistency between the two reporters, but instead arise from differences in imaging modality and acquisition parameters.

Under spinning-disk confocal imaging, mitochondrial markers show highly consistent behavior. In Figure 1B, 1C, and Figure S1B, both the outer and inner mitochondrial membrane (OMM and IMM) markers TOMM-20::mKate2, TOMM-20::GFP, and ATP-3::GFP remain largely associated with mitochondrial structures following laser injury. Although a modest reduction in fluorescence intensity is observed, likely due to partial redistribution of membrane proteins or photo-perturbation after laser-induced injury, the signal is not abolished, and the mitochondrial scaffold remains clearly detectable. In contrast, mitochondrial matrix markers (mito::GFP) and membrane potential (TMRE) are rapidly lost after wounding (Figure 1A and Figure S1A), which cannot be explained by laser-induced photobleaching alone (as supported by additional analyses

in revised Figure S1C and S1D). Together, these observations led us to propose that laser-induced injury causes acute mitochondrial membrane rupture, characterized by rapid loss of matrix components and membrane potential, while the OMM and IMM do not undergo complete disintegration but instead retain a partially preserved membrane scaffold.

The apparent loss of TOMM-20::mYonghong signal in the original Figure 1D is primarily attributable to a technical artifact associated with HIS-SIM (High Intelligent and Sensitive Structured Illumination Microscopy) imaging and reconstruction. While HIS-SIM provides superior spatial resolution and sensitivity, it also introduces differences in signal scaling and dynamic range. In the original dataset, specific acquisition settings combined with a higher zoomed-in visualization of the wound site led to compression of low-intensity signals, causing the TOMM-20 signal at the wound site to appear markedly reduced. Our re-analysis indicates that this effect reflects imaging-dependent distortion, specifically, signal “clipping” in relatively low intensity regions, rather than a true biological loss of the OMM and IMM markers.

To directly address this issue, we performed side-by-side experimental comparisons across imaging platforms. First, we imaged TOMM-20::mYonghong under spinning-disk conditions following laser injury and observed behavior identical to other OMM markers (revised Figure S1F, see also below), thereby excluding fluorophore-specific effects. We then re-acquired and reprocessed the HIS-SIM data using optimized parameters, including improved dynamic range handling (Huang et al., 2018) and sparse deconvolution (Zhao et al., 2021). Under these conditions, TOMM-20::mYonghong clearly retains a membrane-associated signal at the injury site (revised Figure 1D, see also below), consistent with the spinning-disk observations.

Importantly, we also observe that the inner membrane marker ATP-3::GFP extends beyond the TOMM-20 boundary following injury (Figure 1D). This spatial redistribution indicates that mitochondrial membrane integrity may be compromised, even though the OMM scaffold remains partially preserved.

Together, these results support a unified model in which laser-induced injury triggers rapid mitochondrial membrane permeabilization, leading to immediate loss of matrix contents and membrane potential, while the structural membrane scaffold remains partially intact at early time

points. This data reconciles the apparent discrepancies and provides a consistent interpretation across imaging modalities.

Finally, to improve precision and avoid overinterpretation, we have replaced the term “membrane rupture” with “mitochondrial membrane permeabilization” or “severe mitochondrial membrane damage” throughout the manuscript. We are grateful to the reviewer for these constructive comments, which have significantly strengthened the rigor and clarity of our study.

ATP-3::GFP;TOMM-20::mYonghong

Figure S1F. Spinning-disk confocal images of ATP-3::GFP and TOMM-20::mYonghong before and after laser wounding.

ATP-3::GFP;TOMM-20::mYonghong

Figure 1D. HIS-SIM images of ATP-3::GFP and TOMM-20::mYonghong before and after laser wounding. Laser injury induces severe mitochondrial membrane damage but retains the

membrane signals. Arrows indicate disruption of TOMM-20::mYonghong and ATP-3::GFP signals.

Figure 1 and S1 do not rigorously address the issue of photobleaching vs rupture of the mitochondrial membranes, which is a foundational distinction for this study. Fig. S1A show differential loss of matrix vs OM or IM GFPs, but it cannot be ruled out that the different GFPs, due to their fusion to partners, have different sensitivities to laser light. In the matrix experiments, can it be shown that the surrounding region (outside the mitochondria) gains fluorescence upon wounding, as would be expected for rupture of mitochondria and leakage (vs photobleaching) of GFP? With the current results, which rely solely on imaging of fluorescent proteins, it is uncomfortable to conclude that their laser treatment results in membrane rupture.

Response:

We thank the reviewer for this important and thoughtful comment. We fully agree that distinguishing photobleaching from mitochondrial membrane rupture is essential for interpreting our results.

We acknowledge that different GFP fusion proteins may exhibit variable sensitivities to laser illumination; therefore, differential signal loss alone (Figure S1A) is not sufficient to distinguish photobleaching from membrane damage. We appreciate the reviewer's suggestion to directly examine fluorescence redistribution, which provides a more definitive and mechanistically informative test.

To directly address this issue, we quantified fluorescence redistribution after laser injury. Specifically, we measured GFP intensity in the cytosolic regions surrounding mitochondria before and after wounding. We observed a rapid and spatially restricted increase in cytosolic fluorescence signal at the injury site in mito::GFP animals. In contrast, regions distal to the injury site showed no detectable change. This localized gain in cytosolic signal is consistent with the release of matrix-localized GFP upon membrane permeabilization and is not expected under photobleaching, which would instead produce a decrease in fluorescence without redistribution. As an additional control, we examined animals expressing freely diffusible cytosolic GFP. Upon laser injury, these animals exhibited a decrease in fluorescence intensity, likely due to

photobleaching, and did not exhibit any local increase, further distinguishing photobleaching effects from the redistribution observed with mito::GFP.

Together, these results provide direct evidence that the observed signal changes reflect mitochondrial membrane permeabilization rather than differential photobleaching of fluorophores. We have incorporated these quantitative analyses and clarified the interpretation in the revised manuscript (see revised Figure S1C and S1D).

We thank the reviewer again for this valuable suggestion, which has strengthened the rigor and interpretability of our study.

Figure S1C. Wounding induces the release of mito::GFP into the cytosol at the injury site.

Figure S1D. Wounding causes localized photobleaching of freely diffusing GFP at the wound site.

It seems that MAI-1 translocates to mitochondria upon various stresses, but it seem premature to conclude that membrane rupture is the relevant parameter. The authors assume that the various stresses used (cold, genetic mutations) result in rupture.

Response:

We thank the reviewer for this important point and agree that it would be premature to conclude that all stress conditions used in this study necessarily induce mitochondrial membrane rupture.

In the revised manuscript, we have clarified and moderated our interpretation accordingly. We no longer assume that conditions such as cold exposure or genetic perturbations directly cause membrane rupture. Instead, we describe these conditions more cautiously as inducing severe mitochondrial membrane damage that is associated with MAI-1 recruitment.

Importantly, our conclusion that MAI-1 can serve as a reporter of mitochondrial membrane rupture is based primarily on experiments with laser-induced injury, where we provide supporting evidence. This includes rapid irreversible loss of matrix markers and membrane potential, redistribution of matrix-localized GFP into the cytosol, while the overall structural membrane scaffold remains partially preserved. These features are consistent with acute mitochondrial membrane permeabilization.

In contrast, for other stress conditions, we now interpret MAI-1 recruitment more conservatively as an indicator of severe mitochondrial membrane damage that involves membrane compromise, without asserting rupture per se. Supporting this distinction, we show that milder perturbations (e.g., loss of membrane potential or defects in mitochondrial dynamics) are insufficient to trigger MAI-1 recruitment, suggesting that MAI-1 selectively responds to more severe forms of mitochondrial damage.

We have revised the manuscript throughout to clearly distinguish between conclusions supported by direct experimental evidence (laser-induced membrane permeabilization) and those based on correlative observations (other stress conditions), and to ensure consistent and precise terminology. We are grateful to the reviewer for highlighting this important issue, which has strengthened the conceptual clarity of our study.

The authors assume that MAI-1 functions as a reporter of ruptured mitochondria, and maybe even as a facilitator of subsequent mitochondrial degradation. The known function of IF1 is to inhibit the hydrolytic activity of ATP synthase. A simpler model is that, in nematodes, MAI-1 functions to shut down ATP synthase upon membrane rupture. The translocation of cytosolic MAI-1 to mitochondria is simply a consequence of its ability to access its binding partner, which is embedded in the IM and is normally inaccessible. Ruptured mitochondria cannot effectively

provide ATP generation for the cell, and MAI-1 may function to inhibit ATP synthase under such pathological conditions. The subsequent mitochondrial degradation is probably taken care of by other machinery and may have nothing to do with MAI-1. This can be tested by examining mitochondrial degradation in an MAI-1 knockdown or mutant, but the authors argue that this is outside the scope of the study.

Response:

We thank the reviewer for this insightful and constructive suggestion. We agree that the proposed model, where MAI-1 functions to inhibit ATP synthase upon mitochondrial membrane damage, is both plausible and consistent with the established biochemical activity of IF1 family proteins. In this scenario, the observed translocation of cytosolic MAI-1 to damaged mitochondria could indeed reflect newly gained access to its binding partner in the inner mitochondrial membrane following membrane damage.

Importantly, our current study is designed to establish cytosolic IF1 (MAI-1) as a sensitive and rapid reporter of severe mitochondrial membrane damage *in vivo*, rather than to define its endogenous physiological function. We did not intend to conclude that MAI-1 actively facilitates mitochondrial degradation, and we agree that downstream clearance of damaged mitochondria is likely mediated by another mitochondrial quality control pathway.

We also agree that examining mitochondrial degradation in *mai-1* loss-of-function conditions would be a critical experiment to evaluate a potential functional role. However, this analysis is complicated by the presence of two IF1 homologs in *C. elegans* (MAI-1 and MAI-2), which may have overlapping or compensatory functions. Dissecting their respective roles in mitochondrial physiology and stress responses will require substantial additional genetic and functional studies.

Given that this manuscript is presented in a *Tools* format, our primary goal is to introduce and validate MAI-1 as a probe for detecting severe mitochondrial membrane damage. Elucidating its physiological role, including potential regulation of ATP synthase activity under damage conditions, represents an important and interesting direction for future investigation.

We have revised the manuscript to clarify this point and to avoid overinterpretation of MAI-1 function.

The logic for the following conclusion is unclear: "Notably, although endogenous MTS-lacking IF1 proteins have not been identified in mammals, engineered cytosolic IF1 variants from diverse species display conserved rupture-induced mitochondrial recruitment, suggesting that membrane structural damage surveillance is an evolutionarily conserved property of IF1-family proteins. This raises the possibility that analogous mechanisms operate in mammalian tissues, where mitochondrial membrane rupture contributes to neurodegeneration, ischemia-reperfusion injury, and inflammatory pathologies." In fact, the authors' data clearly argue against such a mechanism in mammals or most organisms. They show that such a mechanism is plausible in nematodes (with the caveat in the paragraph above), but not used in mammals.

Response:

We thank the reviewer for this important comment and agree that our original wording overstated the evolutionary implications of our findings. In particular, we acknowledge that endogenous IF1 proteins lacking a mitochondrial targeting sequence have not been identified in mammals, and our data do not support the existence of a native cytosolic IF1-based damage-sensing mechanism in these systems.

We agree with the reviewer that our results are most directly relevant to *C. elegans*, and that extrapolation to mammals or other organisms should be made with caution. In the revised manuscript, we have therefore removed statements implying evolutionary conservation of a damage-sensing mechanism. Instead, we now emphasize a more limited and mechanistically conclusion: IF1-family proteins possess an intrinsic biochemical capacity to associate with mitochondria when inner membrane components become exposed or accessible. This property can be revealed experimentally when these proteins are present in the cytosol.

Accordingly, we have revised the text to avoid suggesting evolutionary conservation of a damage-sensing mechanism and instead highlight that this conserved binding property can be useful *as a tool* to detect severe mitochondrial membrane damage.

The revised text in the manuscript now reads:

“Notably, endogenous MTS-lacking IF1 proteins have not been identified in mammals. However, engineered cytosolic IF1 variants, including human IF1, display damage-induced mitochondrial recruitment, suggesting that IF1 family proteins possess an intrinsic capacity to associate with mitochondria when inner membrane components become accessible. Whether such a mechanism operates under physiological conditions in mammalian cells remains unclear. In this context, MAI-1 provides a useful experimental tool for probing mitochondrial membrane integrity and its consequences in higher organisms.”

We believe this revision more accurately reflects the scope of our data and avoids unsupported evolutionary conclusions.

Since the authors focus on the potential technological utility of the MAI-1 rather than its biological function, they should elaborate on the utility of designing a reporter for catastrophic mitochondrial rupture. The study suggests that many types of rather severe mitochondrial stress, including loss of membrane potential or various genetic mutants, do not recruit MAI-1. MAI-1 recruitment may be limited to very harsh stresses that likely result in broad cellular damage. It would be helpful to point out what specific types of studies would be facilitated by such a reporter.

Response:

We thank the reviewer for this important comment and agree that the utility of MAI-1 as a reporter should be more clearly articulated.

We have revised the manuscript to better highlight these points and clarify the specific experimental contexts in which this reporter provides unique advantages in the discussion part: “The identification of MAI-1 as a damage-specific sensor provides a powerful tool enabling studies not accessible with existing mitochondrial probes. First, MAI-1 allows direct visualization of irreversible mitochondrial membrane damage in vivo, distinguishing reversible dysfunction (e.g., depolarization) from terminal structural failure at single-organelle resolution. Second, MAI-1 supports real-time tracking of damaged mitochondria, facilitating analysis of

their recognition and clearance by quality control pathways. Third, it enables genetic and chemical screens to identify factors that preserve membrane integrity. Forth, MAI-1 provides a quantitative readout of mitochondrial toxicity induced by physiological or pharmacological stress. Finally, its rapid kinetics make it well suited for studying acute injury paradigms requiring high temporal resolution.”

Reviewer #2 (Comments to the Authors (Required)):

The authors have answered my questions and I suggest acceptance.

Response:

We sincerely thank the reviewer for the positive evaluation and helpful feedback. We are pleased that our revisions have addressed the concerns, and we greatly appreciate the reviewer’s support for acceptance of our manuscript.

Reviewer #3 (Comments to the Authors (Required)):

In the revised manuscript, the authors addressed some of my comments and those of other reviewers. However, the authors have not performed fundamental analyses using the mai-1 knockout mutant to demonstrate the physiological function of MAI-1 in the removal of damaged mitochondria. As previously pointed out in Major Comment 1, a detailed analysis of MAI-1 function in this process is considered essential for this study.

Response:

We thank the reviewer for this important and constructive comment and fully agree that defining the physiological function of MAI-1 in mitochondrial quality control is an important question.

However, we would like to clarify that the primary objective of this study is to establish MAI-1 as a sensitive and rapid in vivo reporter of mitochondrial membrane rupture, rather than to determine its endogenous functional role in mitochondrial clearance. We did not intend to

conclude that MAI-1 actively mediates the removal of damaged mitochondria, and we agree that this process is likely governed by established mitochondrial quality control pathways.

We also agree that analyzing mitochondrial degradation in *mai-1* loss-of-function mutants would be an important approach to assess a potential functional role. However, such analysis is not straightforward, as *C. elegans* encodes two IF1 homologs (MAI-1 and MAI-2), which may have partially redundant or compensatory functions. Dissecting their respective contributions would therefore require combinatorial genetic approaches and extensive functional characterization beyond the scope of the current study.

Importantly, we have been careful to avoid overinterpretation and now explicitly state in the revised manuscript that MAI-1 is used here as a reporter, and that its potential physiological role, including in mitochondrial turnover, remains to be determined.

Given that this work is presented in a *Tools* format, we believe that establishing the specificity, sensitivity, and utility of MAI-1 as a probe represents a complete and appropriate contribution. Elucidating its functional role in mitochondrial homeostasis and damage responses will be an important direction for future studies.

April 23, 2026

RE: JCB Manuscript #202511088RR

Suhong Xu
Zhejiang University

Dear Prof. Xu:

Thank you for submitting your revised manuscript entitled "A cytosolic IF1 enables real-time visualization of severe mitochondrial membrane damage". We appreciate your final revisions and responses to ensure the method is reliable and user-friendly for the field. Therefore we would be happy to publish your paper in JCB pending final revisions necessary to meet our formatting guidelines (see details below).

A. MANUSCRIPT ORGANIZATION AND FORMATTING:

1) Text limits: Character count for Tools is < 40,000, not including spaces. Count includes abstract, introduction, results, discussion, and acknowledgments. Count does not include title page, figure legends, materials and methods, references, tables, or supplemental legends.

2) Figures limits: Tools may have up to 10 main text figures.

3) Figure formatting: Scale bars must be present on all microscopy images, including inset magnifications. Molecular weight or nucleic acid size markers must be included on all gel electrophoresis. Aspect ratios of images may not be altered.

4) Statistical analysis: Error bars on graphic representations of numerical data must be clearly described in the figure legend. The number of independent data points (n) represented in a graph must be indicated in the legend. Statistical methods should be explained in full in the materials and methods. For figures presenting pooled data the statistical measure should be defined in the figure legends. Please also be sure to indicate the statistical tests used in each of your experiments (either in the figure legend itself or in a separate methods section) as well as the parameters of the test (for example, if you ran a t-test, please indicate if it was one- or two-sided, etc.). Also, if you used parametric tests, please indicate if the data distribution was tested for normality (and if so, how). If not, you must state something to the effect that "Data distribution was assumed to be normal but this was not formally tested."

5) Abstract and title: The abstract should be no longer than 160 words and should communicate the significance of the paper for a general audience. The title should be less than 100 characters including spaces. Make the title concise but accessible to a general readership.

**** We suggest the following edited title**

"A cytosolic IF1 reporter enables real-time visualization of severe mitochondrial membrane damage"

6) *Materials and methods: Should be comprehensive and not simply reference a previous publication for details on how an experiment was performed. Please provide full descriptions in the text for readers who may not have access to referenced manuscripts.

7) * All antibodies, cell lines, animals, and tools used in the manuscript should be described in full, including accession numbers for materials available in a public repository such as the Resource Identification Portal. Please be sure to provide the sequences for all of your primers/oligos and RNAi constructs in the materials and methods. You must also indicate in the methods the source, species, and catalog numbers (where appropriate) for all of your antibodies. Please also indicate the acquisition and quantification methods for immunoblotting/western blots.

8) Microscope image acquisition: The following information must be provided about the acquisition and processing of images:

- a. Make and model of microscope
- b. Type, magnification, and numerical aperture of the objective lenses
- c. Temperature
- d. Imaging medium

- e. Fluorochromes
- f. Camera make and model
- g. Acquisition software
- h. Any software used for image processing subsequent to data acquisition. Please include details and types of operations involved (e.g., type of deconvolution, 3D reconstitutions, surface or volume rendering, gamma adjustments, etc.).

10) Supplemental materials: There are strict limits on the allowable amount of supplemental data. Articles may have up to 5 supplemental figures. Please also note that tables, like figures, should be provided as individual, editable files. A summary of all supplemental material should appear at the end of the Materials and methods section.

13) ORCID IDs: ORCID IDs are unique identifiers allowing researchers to create a record of their various scholarly contributions in a single place. Please note that ORCID IDs are now **required** for all authors. At resubmission of your final files, please be sure to provide your ORCID ID and those of all co-authors.

Please note that JCB now requires authors to submit Source Data used to generate figures containing gels and Western blots with all revised manuscripts. This Source Data consists of fully uncropped and unprocessed images for each gel/blot displayed in the main and supplemental figures. For assays performed using capillary electrophoresis and/or immunoassay-based detection, authors should instead provide the electropherogram graph(s) for each experiment, plotting fluorescence/chemiluminescence intensity vs. molecular weight/size. Please be sure to provide one Source Data file for each figure gels, blots, and/or capillary electrophoresis assays along with your revised manuscript files. File names for Source Data figures should be alphanumeric without any spaces or special characters (i.e., SourceDataF#, where F# refers to the associated main figure number or SourceDataFS# for those associated with Supplementary figures). For traditional gels and blots, the lanes of the gels/blots should be labeled as they are in the associated figure, the place where cropping was applied should be marked (with a box), and molecular weight/size standards should be labeled wherever possible. For capillary electrophoresis assays, each trace in the graph should be color-coded and labeled to indicate which protein, gene, or sample is being measured (please try to avoid red/green combinations to accommodate our color-blind readers).

Journal of Cell Biology now requires a data availability statement for all research article submissions. These statements will be published in the article directly above the Acknowledgments. The statement should address all data underlying the research presented in the manuscript. Please visit the JCB instructions for authors for guidelines and examples of statements at (<https://rupress.org/jcb/pages/editorial-policies#data-availability-statement>).

B. FINAL FILES:

The license to publish form must be signed before your manuscript can be sent to production. A link to the license to publish form will be sent to the corresponding author only. Please take a moment to check your funder requirements before choosing the appropriate license.

Thank you for your attention to these final processing requirements. Please revise and format the manuscript and upload materials within 14 days. If you need an extension for whatever reason, please let us know and we can work with you to determine a suitable revision period.

Thank you for this interesting contribution, we look forward to publishing your paper in Journal of Cell Biology.

Sincerely,

Hong Zhang, PhD
Monitoring Editor

Andrea L. Marat, PhD
Deputy Editor

Journal of Cell Biology

Reviewer #1 (Comments to the Authors (Required)):

My concerns have been addressed.